# ANESSUITE: A COMPREHENSIVE BENCHMARK AND DATASET SUITE FOR ANESTHESIOLOGY REASONING IN LLMS

**Xiang Feng**[1]* **Wentao Jiang**[1]* **Zengmao Wang**[1] **Yong Luo**[1]†
**Pingbo Xu**[2,3] **Baosheng Yu**[4,5] **Hua Jin**[6,7] **Jing Zhang**[1]†

[1] School of Computer Science, National Engineering Research Center for Multimedia Software and Hubei Key Laboratory of Multimedia and Network Communication Engineering, Wuhan University, China
[2] Department of Anesthesiology, Zhejiang Cancer Hospital, China
[3] Institute of Medicine, Chinese Academy of Sciences, Hangzhou, Zhejiang, China
[4] Lee Kong Chian School of Medicine, Nanyang Technological University, Singapore
[5] Centre of AI in Medicine, Nanyang Technological University, Singapore
[6] Department of Anesthesiology, First People's Hospital of Yunnan Province, China
[7] Kunming University of Science and Technology, China
{fengxiang_cs, jiang_wentao, wangzengmao, luoyong}@whu.edu.cn
xupb@zjcc.org.cn, baosheng.yu@ntu.edu.sg, jinhuakm@163.com
jingzhang.cv@gmail.com

## ABSTRACT

The application of large language models (LLMs) in the medical field has garnered significant attention, yet their reasoning capabilities in more specialized domains like anesthesiology remain underexplored. To bridge this gap, we introduce AnesSuite, the first comprehensive dataset suite specifically designed for anesthesiology reasoning in LLMs. The suite features AnesBench, an evaluation benchmark tailored to assess anesthesiology-related reasoning across three levels: factual retrieval (System 1), hybrid reasoning (System 1.x), and complex decision-making (System 2). Alongside this benchmark, the suite includes three training datasets that provide an infrastructure for continued pre-training (CPT), supervised fine-tuning (SFT), and reinforcement learning with verifiable rewards (RLVR). Leveraging this suite, we develop Morpheus, the first baseline model collection for anesthesiology reasoning. Despite undergoing limited training with SFT and group relative policy optimization (GRPO), Morpheus not only achieves substantial improvements in anesthesiology that rival larger-scale models, but also demonstrates enhanced reasoning capabilities across general medical and broad-domain benchmarks. Furthermore, through comprehensive evaluations and experiments, we analyze the key factors influencing anesthesiology reasoning performance, including model characteristics, training strategies and training data. Both AnesSuite and Morpheus will be open-sourced at https://github.com/MiliLab/AnesSuite.

## 1 INTRODUCTION

The success of LLMs has fundamentally reshaped the domain of medical artificial intelligence and catalyzed the creation of specialized medical models (Liu et al., 2024b). However, the reasoning ability of LLMs within medicine and its subdisciplines still need significant improvement, particularly when compared with general reasoning tasks such as mathematics and programming (Weng et al., 2023).

---

*Equal Contribution
†Corresponding Author

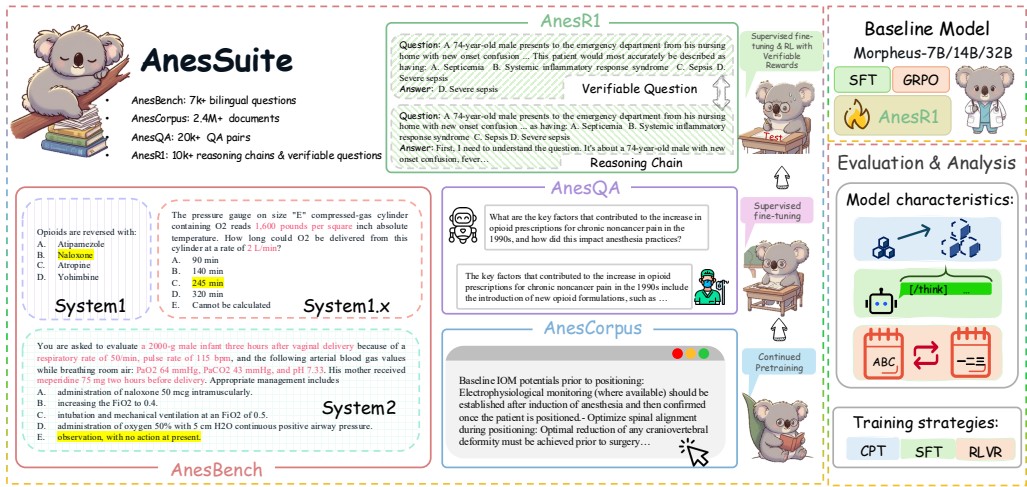

Figure 1: **Overview of AnesSuite.** AnesSuite is composed of four components: *AnesBench*, a cross-lingual structured benchmark; *AnesCorpus*, a collection of anesthesiology documents; *AnesQA*, a question-answering dataset derived from domain literature; and *AnesR1*, a dataset featuring verifiable anesthesiology questions with chain-of-thought annotations. Leveraging this suite, we developed *Morpheus*, the first collection of reasoning LLMs for anesthesiology. Subsequent ablation and experiments identified key factors influencing reasoning performance in this specialized domain.

Anesthesiology, as a highly specialized discipline, requires extensive medical expertise and high-stakes clinical decision-making. In clinical practice, anesthesiology encompasses the simultaneous management of diverse and interdependent physiological systems—including airway and respiratory function, cardiovascular stability, electrolyte balance and sedation levels, as well as perioperative physiological fluctuations. This multifaceted responsibility means that anesthetic decision-making requires not only the factual recall of System 1 thinking but also the complex, context-aware judgment and higher-order reasoning of System 2 (Kahneman, 2011). Existing medical benchmarks and models often overlook the unique characteristics of anesthesiology (Panagoulias et al., 2024; Olatunji et al., 2024; Abacha et al., 2019; Liu et al., 2024c). Others focus mainly on assessing factual recall (Li et al., 2023; Grzybowski et al., 2024; Wang et al., 2024a; Zhou et al., 2025).

To address this gap, we introduce the first comprehensive suite of benchmark and datasets, **Anes-Suite**, dedicated to anesthesiology reasoning. This suite includes **AnesBench**, a bilingual and structured benchmark comprising 7,972 questions designed to evaluate both factual retrieval and clinical decision-making in anesthesiology, which accompanied by a full range of training resources: **AnesCorpus**, an extensive collection of over 2.4 million documents for continued pre-training; **AnesQA**, a SFT (Ouyang et al., 2022) dataset with over 20,713 question-answer (QA) pairs; **AnesR1**, a dataset with over 10,287 instances of verifiable anesthesiology questions with chain-of-thought annotations, which can be used for SFT (Ouyang et al., 2022) or RLVR (Lambert et al., 2024).

Leveraging AnesSuite, we propose Morpheus, the first collection of reasoning LLMs specifically designed for anesthesiology. Initialized from the Qwen2.5-7B, Qwen2.5-14B and Qwen2.5-32B (Yang et al., 2024), it was trained on AnesR1 using SFT and GRPO (Shao et al., 2024). Despite limited training, Morpheus consistently enhances reasoning performance across domain-specific, general medical, and general-domain benchmarks, rivaling the performance of larger-scale models.

Moreover, using AnesBench, we evaluate a range of leading LLMs, including proprietary models such as GPT-4o (OpenAI, 2024a), Claude-3.7-Sonnet (Anthropic, 2025), Gemini-2.5-Flash and Gemini-2.5-Pro (Comanici et al., 2025). We also conducted extensive ablation and analytical experiments using the training datasets in AnesSuite. Our analysis yields several key insights: (1) Model performance benefits from larger model scales, whereas System 2 tasks exhibit comparatively smaller gains. Both systems experience diminishing marginal returns, with performance improvements decreasing as model scale increases. (2) The length of the Chain-of-Thought (Wei et al., 2022, CoT) is a critical factor in enhancing performance, particularly for System 2 tasks, as it

enables more structured, step-by-step reasoning. (3) Language transferability remains a significant factor limiting the performance of multilingual models. (4) Model's multilingual anesthesiology reasoning capabilities are highly sensitive to the training data of CPT phase, necessitating careful corpus management. (5) Data from the general medical domain serves as a valuable complement to anesthesiology-specific data, enhancing overall reasoning performance.

Our contributions are threefold: (1) Dataset Suite Construction: We introduce AnesSuite, the first comprehensive dataset suite for anesthesiology reasoning. It includes AnesBench, a structured, cross-lingual benchmark for evaluating factual retrieval and clinical decision-making, along with three training datasets supporting CPT, SFT, and RLVR. (2) Baseline Model: We present Morpheus, the first collection of reasoning LLMs for anesthesiology. With SFT and GRPO training on the AnesR1 dataset, Morpheus yields consistent reasoning enhancements across domain-specific, general medical, and general-domain benchmarks, underscoring the efficacy of data from reasoning-intensive disciplines like anesthesiology. (3) Comprehensive Evaluation and Analysis: Through systematic evaluations and extensive ablation studies, we provide an in-depth analysis of factors influencing anesthesiology reasoning, such as model characteristics, training strategies and training data. Overall, we are committed to leveraging our dataset suite, baseline models, and in-depth analysis collectively provide valuable insights for developing LLMs with more enhanced anesthesiology reasoning capabilities. We will publicly release AnesSuite, Morpheus, and all associated resources.

## 2 RELATED WORK

### 2.1 REASONING LLMs

Reasoning refers to solving problems involve complex, multi-step processes with intermediate steps (Li et al., 2025b). The emergence of reasoning LLMs such as OpenAI o1/o3 (OpenAI, 2024c; 2025) and Deepseek R1 (Guo et al., 2025) has spurred interest in enhancing LLM's reasoning ability. Recent research indicates that LLMs tend to generate longer outputs when solving mathematical or reasoning problems (Xiang et al., 2025). Model performance benefits from scaling the length of the reasoning process (Tang et al., 2025; Huang et al., 2025). Models that exhibit strong exploratory tendencies or incorporate intrinsic verification mechanisms in their outputs often demonstrate enhanced reasoning capabilities (Xiang et al., 2025; Guo et al., 2025). Moreover, different model scales have been demonstrated to exhibit significant differences in training dynamics and reasoning characteristics (OpenAI, 2024c; 2025; Zhao et al., 2024). On the other hand, training and inference paradigms, such as reinforcement learning fine-tuning (Luong et al., 2024; Guo et al., 2025; Team et al., 2025b, RFT), self-improvement methods (Zelikman et al., 2022; Gulcehre et al., 2023; Singh et al., 2024), and structured search techniques exemplified by MCTS (Bi et al., 2025; Qiu et al., 2024; Li et al., 2025a), have shown significant development potential.

### 2.2 ANESTHESIA-RELATED BENCHMARKS AND LLMs

Anesthesiology, due to its unique interdisciplinary nature, is often implicitly classified under surgery, dentistry, or similar categories (Panagoulias et al., 2024; Olatunji et al., 2024; Abacha et al., 2019), even in benchmarks that categorize based on different departments. Some benchmarks explicitly treat anesthesiology as a separate category (Grzybowski et al., 2024; Wang et al., 2024a), while some studies employ existing anesthesiology-related questions (Khan et al., 2024; Angel et al., 2022; Shay et al., 2023; Kinoshita et al., 2024; Patnaik & Hoffmann, 2023) (e.g., exam questions from American Society of Anesthesiologists, ASA and Japanese Society of Anesthesiologists, JSA) to evaluate LLMs. However, the overall scale of these evaluations for anesthesiology remains very limited. Chinese Anesthesiology Benchmark (Zhou et al., 2025, CAB) is the first to focus primarily on anesthesiology. Nevertheless, CAB's primary emphasis on factual retrieval questions results in insufficient exploration of the unique challenges in anesthetic reasoning and decision-making. Additionally, its restriction to the Chinese language further limits its applicability and impact. Moreover, Hypnos and some works (Wang et al., 2024c; Guthrie et al., 2024) attempts to use anesthesiology-related corpora for supervised fine-tuning to improve question-answering abilities, but they have not significantly enhanced the reasoning capabilities.

| Dataset | Data type | Size |
|---------|-----------|------|
| AnesBench | MCQs | 4.4k (EN) + 3.5k (CH) |
| AnesCorpus | Plain Text | 1.8M (EN) + 0.6M (CH) |
| AnesQA | QA pairs | 20k (EN) |
| AnesR1 | MCQs + CoT | 3.2k (EN) + 7k (CH) |

Table 1: Overview of datasets in AnesSuite.

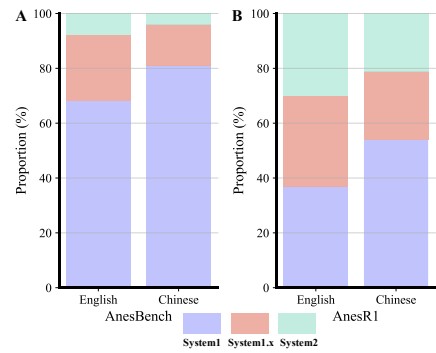

Figure 2: Distribution of cognitive demand levels in AnesBench and AnesR1.

## 3 ANESSUITE DATASET

We propose AnesSuite, the first comprehensive dataset suite specifically designed for reasoning and decision-making in anesthesiology. This suite includes AnesBench, a cross-lingual structured benchmark. In addition, AnesSuite includes three training datasets, AnesCorpus, AnesQA, and AnesR1, which collectively support a range of fine-tuning and application in anesthesiology.

### 3.1 DATASETS OVERVIEW

AnesBench is a bilingual benchmark featuring parallel English and Chinese subsets, containing 4,418 and 3,554 multiple-choice questions (MCQs), respectively. Its core design is a three-level of cognitive demand: System 1, focused on factual recall; System 2, involving complex reasoning; and System 1.x, a hybrid category merging recall with elementary reasoning (Saha et al., 2025). Critically, shown in Fig. 2, questions demanding higher-order cognitive skills (System 1.x and System 2) constitute a substantial 20-30% of the total items. This hierarchical structure and emphasis on complex tasks, can provide a nuanced methodology for evaluating the advanced anesthesiological reasoning capabilities of LLMs.

Complementing the benchmark are three training datasets designed to support model development. AnesCorpus is a large-scale text collection for continued pre-training, comprising over 1.8 million English and 0.6 million Chinese documents pertinent to anesthesiology. AnesQA contains 20,713 English QA pairs, where each question is annotated with one of five question type labels to enable targeted fine-tuning, as presented in Fig. 3. AnesR1 comprises a total of 10,287 items, distributed across Chinese and English subsets. Each item consists of a verifiable MCQ accompanied by a detailed CoT reasoning process. Like AnesBench, it also employs the three-level cognitive demand framework. The inclusion of detailed CoT reasoning is a defining feature of AnesR1, which is reflected in its data structure. As illustrated in Fig. 4, this leads to substantially longer answers in AnesR1 compared to the concise answers in AnesQA, even though question lengths remain comparable across AnesBench, AnesQA, and AnesR1. Collectively, these training datasets provide a foundational infrastructure that supports the entire workflow, from CPT and SFT to RLVR.

### 3.2 DATASETS CURATION

AnesBench was constructed from authoritative sources, including American Board of Anesthesiology (ABA) examination materials, standardized textbooks, and validated online assessment tools. After collection, each item was annotated for its cognitive demand level using the DeepSeek-R1 (Guo et al., 2025). This annotation was performed via a prompting strategy that incorporated comprehensive guidelines and few-shot examples (the complete prompt is included in the Appendix A.2). For quality assurance, 60% of the items were randomly chosen for manual review. Subsequently, we conducted a data contamination (whether the questions have been seen in each model training) analysis using a specialized algorithm (Ni et al., 2025) for MCQs to verify that data

contamination was minimal. The detailed results of the manual verification and the data leakage analysis are presented in the Appendix A.1.

To create AnesCorpus, we sourced data from two large-scale web datasets, namely Fineweb[1] and Chinese Fineweb[2]. We employed a frequency-based approach to filter this extensive collection. Specifically, we curated a list of keywords pertinent to anesthesia and pain management and categorized them into two groups (the complete list is available in the Appendix A.4). A document was included in AnesCorpus only if it met two criteria: (1) at least one keyword from the first group appeared once per 4,000 characters, and (2) the document contained at least one keyword from the second group. To prevent contamination from this pre-training corpus to the AnesBench benchmark, we implemented a two-stage decontamination process. The initial stage used a rapid n-gram-based screening to identify documents with potential content overlap. In the second stage, these flagged documents were subjected to a more granular analysis using Longest Common Substring (LCS). In line with previous studies, such as HuatuoGPT-o1 (Chen et al., 2024) and Med-PaLM 2 (Singhal et al., 2025), documents with an LCS exceeding 64 characters when compared to a question in AnesBench were removed. A detailed description of this decontamination protocol is provided in the Appendix A.4.

The construction of AnesQA began with sourcing anesthesiology-related research papers from PubMed [3]. These papers underwent an extensive cleaning process—including content removal, format standardization, and relevance filtering—to yield high-quality, domain-specific text. We then employed an LLM-based automated pipeline to generate question-answer pairs. This pipeline involved segmenting the cleaned papers into text segments based on length and section structure. For each segment, LLaMA3.3-70B-Instruct (Grattafiori et al., 2024) was utilized to generate a self-contained question. Subsequently, Qwen2.5-72B-Instruct (Yang et al., 2024) was employed to select the most informative and domain-relevant questions and to generate their corresponding answers. The complete prompts used in this process are detailed in the Appendix A.5. While this dual-model approach can partially mitigate the inherent biases of LLMs, the generated content may still contain formatting errors or hallucinations. To address this, we conducted a further manual verification stage: we inspected a sample of the LLM outputs to identify common errors, then applied filters to remove all QA pairs exhibiting these issue.

The construction of AnesR1 followed a procedure similar to that of AnesBench, starting with the collection of anesthesiology multiple-choice questions from a distinct set of authoritative sources. During data deduplication, we implemented a process mirroring the second stage of the AnesCorpus decontamination: any question with an LCS greater than 64 characters when compared to an item in AnesBench was flagged for manual inspection. Questions deemed highly similar were subsequently eliminated. For each validated question, reasoning trajectories were generated using DeepSeek-R1 (Guo et al., 2025). A rejection sampling protocol was then enforced, accepting only those trajec-

| Dataset | Max | Mean |
|---|---|---|
| AnesBench-EN | 9 | 4.3 |
| AnesBench-CH | 7 | 5.0 |
| AnesR1-EN | 5 | 4.5 |
| AnesR1-CH | 8 | 4.9 |

Table 2: Number of choices in AnesBench and AnesR1.

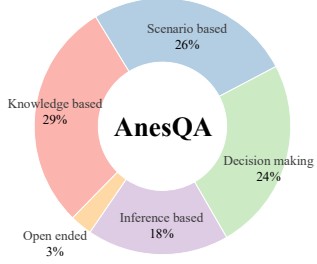

Figure 3: Distribution of AnesQA question type.

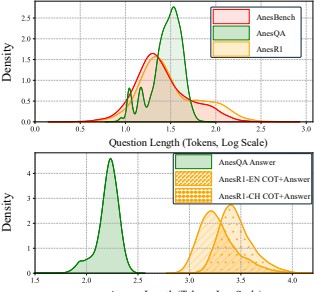

Figure 4: Data length distribution for AnesBench, AnesQA, and AnesR1.

[1]Fineweb: https://huggingface.co/datasets/HuggingFaceFW/fineweb

[2]Chinese Fineweb: https://huggingface.co/datasets/opencsg/chinese-fineweb-edu

[3]PubMed: https://pubmed.ncbi.nlm.nih.gov/

Table 3: **Main Evaluation Results on AnesBench.** The highest and second-highest accuracies in each column are highlighted in bold and underlined, respectively. Complete results are in Appendix B.3.

| Model | AnesBench-English | | | | AnesBench-Chinese | | | | Avg. |
|---|---|---|---|---|---|---|---|---|---|
| | Sys1 | Sys1.x | Sys2 | Total | Sys1 | Sys1.x | Sys2 | Total | |
| **Medical Specific LLMs** | | | | | | | | | |
| BioMistral 7B Chat | 0.43 | 0.30 | 0.32 | 0.39 | 0.24 | 0.25 | 0.16 | 0.24 | 0.31 |
| HuatuoGPT-o1 7B | 0.56 | 0.45 | 0.38 | 0.52 | 0.73 | 0.59 | 0.55 | 0.71 | 0.61 |
| HuatuoGPT-o1 8B | 0.58 | 0.46 | 0.39 | 0.53 | 0.58 | 0.46 | 0.46 | 0.56 | 0.54 |
| HuatuoGPT-o1 70B | 0.70 | 0.58 | 0.48 | 0.65 | 0.71 | 0.57 | 0.49 | 0.68 | 0.66 |
| HuatuoGPT-o1 72B | 0.71 | 0.61 | 0.48 | 0.67 | 0.79 | 0.67 | 0.61 | 0.76 | 0.71 |
| **Open-Source LLMs** | | | | | | | | | |
| Qwen3-0.6B | 0.39 | 0.31 | 0.26 | 0.36 | 0.34 | 0.29 | 0.22 | 0.35 | 0.31 |
| Qwen3-1.7B | 0.48 | 0.37 | 0.30 | 0.44 | 0.53 | 0.39 | 0.34 | 0.50 | 0.50 |
| Qwen3-4B | 0.60 | 0.46 | 0.34 | 0.54 | 0.64 | 0.66 | 0.50 | 0.37 | 0.58 |
| Qwen3-8B | 0.65 | 0.50 | 0.40 | 0.60 | 0.72 | 0.44 | 0.39 | 0.70 | 0.65 |
| Qwen3-14B | 0.70 | 0.57 | 0.45 | 0.65 | 0.79 | 0.63 | 0.52 | 0.76 | 0.70 |
| Qwen3-32B | 0.72 | 0.64 | 0.48 | 0.68 | 0.81 | 0.64 | 0.57 | 0.78 | 0.70 |
| Qwen3-30B-A3B | 0.73 | 0.60 | 0.48 | 0.68 | 0.77 | 0.62 | 0.50 | 0.74 | 0.61 |
| Qwen3-235B-A22B | 0.78 | 0.67 | 0.57 | 0.74 | 0.85 | 0.72 | 0.58 | 0.81 | 0.77 |
| Llama-4-Scout-17B-16E | 0.77 | 0.66 | 0.55 | 0.72 | 0.80 | 0.66 | 0.55 | 0.77 | 0.74 |
| Llama-4-Maverick-17B-128E | 0.83 | 0.73 | 0.64 | 0.79 | 0.86 | 0.72 | 0.59 | 0.83 | 0.81 |
| GPT-OSS-20B | 0.70 | 0.60 | 0.52 | 0.66 | 0.71 | 0.58 | 0.49 | 0.68 | 0.67 |
| GPT-OSS-120B | 0.80 | 0.72 | 0.62 | 0.76 | 0.80 | 0.64 | 0.54 | 0.76 | 0.76 |
| DeepSeek-V3 | 0.77 | 0.69 | 0.55 | 0.73 | 0.87 | 0.75 | 0.60 | 0.84 | 0.78 |
| DeepSeek-R1 | 0.85 | 0.78 | 0.70 | 0.82 | 0.86 | **0.77** | **0.61** | 0.83 | 0.82 |
| **Proprietary LLMs** | | | | | | | | | |
| GPT-4o | 0.81 | 0.72 | 0.59 | 0.77 | 0.79 | 0.64 | 0.52 | 0.76 | 0.76 |
| Gemini-2.5-Flash | 0.84 | 0.76 | 0.68 | 0.81 | 0.84 | 0.72 | 0.59 | 0.81 | 0.81 |
| Gemini-2.5-Pro | **0.89** | **0.82** | **0.77** | **0.86** | **0.88** | 0.75 | 0.60 | **0.85** | **0.85** |
| Claude-3.7-Sonnet | 0.80 | 0.73 | 0.63 | 0.77 | 0.82 | 0.65 | 0.55 | 0.78 | 0.77 |

tories that culminated in the correct answer. If three sampling attempts failed to produce a correct solution, the source question was excluded from the final AnesR1 dataset.

# 4 EVALUATION AND BASELINE MODEL

## 4.1 EVALUATION

**Benchmarked LLMs.** We conducted a comprehensive zero-shot evaluation on over 50 LLMs from diverse sets. The evaluation encompassed four leading proprietary models, namely GPT-4o (OpenAI, 2024a), Gemini-2.5-Flash, Gemini-2.5-Pro (Comanici et al., 2025), and Claude-3.7-Sonnet (Anthropic, 2025). The selection of open-source models included both domain-specific LLMs, such as HuatuoGPT-o1 (Chen et al., 2024) and BioMistral (Labrak et al., 2024), as well as prominent general-purpose LLMs like DeepSeek-R1/V3 (Guo et al., 2025; Liu et al., 2024a), Qwen3 (Team, 2025), and Llama4 (Meta, 2025). To facilitate a more granular analysis, we included models of varying parameter scales from each open-source family. While the main text presents a selection of results, the complete list of models and their corresponding evaluation outcomes are detailed in the Appendix B.1.

**Supplementary Evaluation** To further refine our assessment, we introduced a supplementary open-ended subset, which focused on open-ended questions in anesthesiology, to capture and analyze the more nuanced reasoning capabilities of LLMs within the anesthesiology domain. Additionally, we conducted a comparative analysis using the MCQ subset from CAB (Zhou et al., 2025) against AnesBench. The detailed methodologies and results for these supplementary evaluation are available in the Appendix G.

Table 4: Evaluation Results of Morpheus on AnesBench.

| Model | Stage | | AnesBench-English | | | | AnesBench-Chinese | | | | Avg. |
|---|---|---|---|---|---|---|---|---|---|---|---|
| | SFT | GRPO | Sys1 | Sys1.x | Sys2 | Total | Sys1 | Sys1.x | Sys2 | Total | |
| Qwen2.5-7B-Instruct | - | - | 0.56 | 0.44 | 0.36 | 0.51 | 0.69 | 0.55 | 0.55 | 0.66 | 0.59 |
| Morpheus-7B (SFT only) | ✓ | ✗ | 0.59 | 0.45 | 0.35 | 0.54 | 0.58 | 0.42 | 0.47 | 0.56 | 0.54 |
| Morpheus-7B | ✓ | ✓ | 0.61 | 0.48 | 0.40 | 0.56 | 0.73 | 0.55 | 0.50 | 0.70 | 0.63 |
| Qwen2.5-14B-Instruct | - | - | 0.61 | 0.52 | 0.41 | 0.57 | 0.75 | 0.63 | 0.53 | 0.72 | 0.64 |
| Morpheus-14B (SFT only) | ✓ | ✗ | 0.65 | 0.51 | 0.38 | 0.60 | 0.57 | 0.51 | 0.45 | 0.55 | 0.57 |
| Morpheus-14B | ✓ | ✓ | 0.68 | 0.55 | 0.47 | 0.63 | 0.78 | 0.62 | 0.53 | 0.75 | 0.69 |
| Qwen2.5-32B-Instruct | - | - | 0.65 | 0.55 | 0.44 | 0.61 | 0.79 | 0.65 | 0.54 | 0.76 | 0.68 |
| Morpheus-32B (SFT only) | ✓ | ✗ | 0.72 | 0.60 | 0.52 | 0.67 | 0.67 | 0.53 | 0.47 | 0.64 | 0.65 |
| Morpheus-32B | ✓ | ✓ | 0.72 | 0.61 | 0.54 | 0.68 | 0.80 | 0.66 | 0.53 | 0.77 | 0.72 |
| Qwen2.5-72B-Instruct | - | - | 0.69 | 0.58 | 0.49 | 0.65 | 0.79 | 0.66 | 0.52 | 0.76 | 0.70 |

**Implementation details.** The metric reported is Accuracy on multiple-choice questions. To ensure reproducibility, all models were evaluated with the temperature set to 0 and maximum output length set to 2048 tokens. We employed a Zero-Shot CoT prompting strategy to encourage structured, step-by-step reasoning (Wei et al., 2022). A detailed description of the experimental setting and the prompt is provided in Appendix B.2.

**Main Results.** Tab. 3 presents the main evaluation results on the AnesBench. We observe that complex anesthesiology reasoning remains a significant challenge for contemporary LLMs. On the AnesBench, most open-source models achieve scores below 0.5 on System 2 questions. Even proprietary models, such as GPT-4o and Claude-3.7-Sonnet, only achieve accuracies of around 0.5 to 0.6. Furthermore, a consistent trend across most models is that performance on System 1.x and System 2 tasks is markedly lower than on System 1 tasks. This discrepancy suggests that the principal challenge for current LLMs is not a simple deficit in anesthesiological knowledge, but rather an underdeveloped ability to apply this knowledge to complex reasoning problems. A major factor contributing to these lower scores in complex reasoning tasks is the prevalence of logical hallucinations during the step-by-step generation process. To better understand this phenomenon, we provide a detailed evaluation of reasoning hallucinations, such as Non-Sequiturs and Over-extrapolations, in Appendix G.3. Additionally, medical specific LLMs do not demonstrate a significant advantage over recent general-purpose reasoning models in our evaluation. This finding underscores the specialized nature of anesthesiological reasoning, distinguishing it from general medical domain.

## 4.2 BASELINE MODEL

We developed baseline models by fine-tuning Qwen2.5-7B, Qwen2.5-14B and Qwen2.5-32B (Yang et al., 2024) on the AnesR1. Both models underwent SFT followed by GRPO and are designated Morpheus-7B, Morpheus-14B and Morpheus-32B, respectively. In alignment with established paradigms, the SFT stage serves as a cold-start initializing for the subsequent GRPO training. For comparative purposes, we also present these intermediate models, which obtained after the limited SFT steps. Complete training details are provided in Appendix C.2. In addition, the results of AnesR1 on more models can be found in Tab. 11.

The evaluation results for Morpheus are presented in Tab. 4. A key observation is that, after GRPO training, the model achieves consistent improvements across various systems, demonstrating the effectiveness of the GRPO algorithm in enhancing reasoning capabilities. Notably, following SFT and GRPO training on AnesR1, Morpheus-7B, Morpheus-14B and Morpheus-32B reach performance levels comparable to Qwen2.5-14B-Instruct, Qwen2.5-32B-Instruct and Qwen2.5-72B-Instruct, respectively. This highlights the efficacy of AnesR1 in cultivating complex reasoning within the specialized domain of anesthesiology. Furthermore, as shown in Tab 18 (Appendix G.1), Morpheus ex-

hibits enhanced reasoning capabilities across a wide range of general medical and general-domain benchmarks (e.g., MMLU, MedQA). However, on specific subsets of these benchmarks, the improvements exhibited by Morpheus are less pronounced. This may be associated with the correlation between the generalization of reasoning capabilities acquired through reinforcement learning and the extent of data exposure during the pre-training phase, as discussed by (Cheng et al., 2025). Furthermore, considering that AnesR1 focuses exclusively on anesthesiology, a domain typically underrepresented in standard LLM training corpora, these findings may offer valuable insights into another pertinent question: whether reasoning skills acquired via reinforcement learning in sparsely exposed domains can effectively transfer to more general domains.

## 5 ABLATION AND ANALYSIS

In this section, we present a series of ablation studies and analyses to provide insights for developing advanced LLMs for anesthesiology reasoning. Our investigation examines two key aspects: model characteristics, training methodologies and data. Furthermore, the majority of the insights and claims presented in this section have undergone statistical significance testing. The full results are provided in Appendix F.

### 5.1 MODEL CHARACTERISTIC

**Performance Across Model Scale.** Models are arranged in increasing order of their accuracy on the AnesBench-English subset. As depicted in Fig. 5, there is a strong positive correlation between model performance and scale, which exhibits diminishing marginal returns; larger increases in model scale correspond to progressively smaller performance gains. Furthermore, the fitted-line slopes for System1 and System1.x are relatively close. In contrast, the slope for System2 is significantly lower than System1.x, indicating that performance gains from increasing model size are markedly smaller for System2.

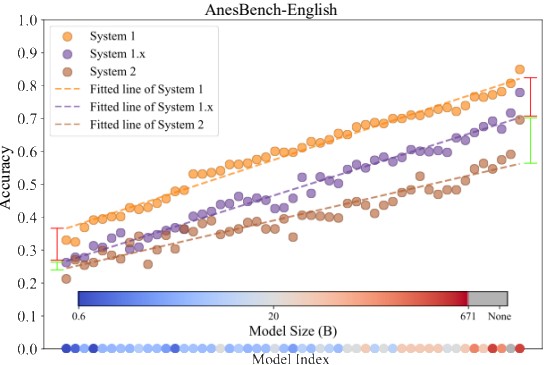

Figure 5: **Impact of Model Scale.** Colors denote model scale (model index) and problem type (scatter points). Models are sorted by overall score in ascending order.

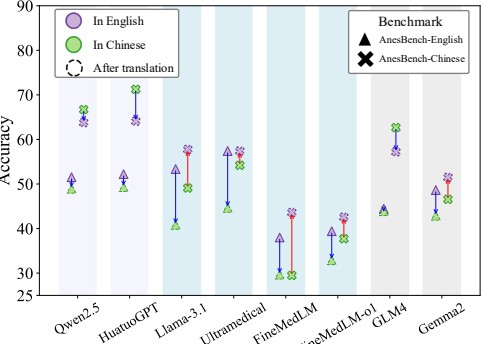

Figure 6: **Multilingual Assessment on Translated Benchmarks.** Background colors denote base models: purple for Qwen2.5-7B, blue for Llama-3.1-8B, gray for others.

**Performance Across Language.** To evaluate model performance across different linguistic settings, we performed a cross-translation of the AnesBench-English and AnesBench-Chinese subsets using GPT-4o (OpenAI, 2024a). Specifically, the English subset was translated into Chinese, and the Chinese subset into English. This process resulted in each subset having versions in both languages, enabling direct cross-lingual comparisons. The prompts and details are detailed in Appendix C.1.

As shown in Fig. 6, Qwen2.5-7B-based and other models exhibit minimal disparity between English and Chinese performance. In contrast, Llama-3.1-8B-based models, despite claiming multilingual support (Grattafiori et al., 2024), perform notably worse in Chinese. These results indicate that language transferability remains a key factor influencing multilingual model performance. According to the theory proposed in (Chen et al., 2025), once an LLM reaches a stable language learning stage,

different languages form independent knowledge systems rather than relying on translation. At this stage, a lack of domain-specific knowledge within the language-specific knowledge system can lead to substantial performance disparities across languages. Therefore, we recommend supplementing bilingual domain-specific knowledge during the different training stage to mitigate cross-linguistic performance gaps.

**Performance Across Output Length** By comparing output length and accuracy across three levels of questions in AnesBench-English, we discovered: (1) Models with longer CoT reasoning processes tend to exhibit superior response performance. As illustrated in the right subplot of Fig. 7, for System2 questions, models achieve higher scores with longer outputs. (2) However, this trend is not pronounced in System1 and System1.x, which do not require extensive reasoning processes. As shown in the left and middle subplots of Fig. 7, the scores of models are almost solely correlated with the size of the model.

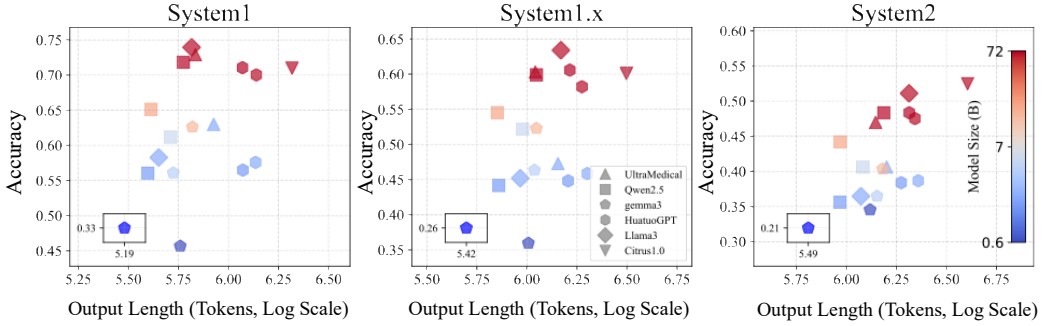

Figure 7: **Impact of Output Length** Shapes and colors denote model families and scales.

## 5.2 TRAINING STRATEGY AND DATA

This section focuses on two key research questions: (1) the effect of CPT training strategies on model performance, and (2) the impact of training data from different domains on the model's performance.

To investigate these questions, we subjected models to SFT for an equivalent number of steps using two distinct datasets: AnesQA, representing the anesthesiology-specific domain, and the bilingual Medical-o1 (Chen et al., 2024), representing the general medical domain. To isolate the impact of CPT, this process was applied to models both with and without prior continued pre-training on AnesCorpus. Notably, since AnesQA lacks native Chinese data, we translated the dataset using GPT-4o-mini (OpenAI, 2024b). This step was crucial to align its format with the setting of Medical-o1 and to ensure the validity of evaluations on AnesBench-Chinese. Further details on the training and translation are provided in Appendix C.3 and Appendix C.1. The results, presented in Tab. 5, yield two key insights.

Table 5: **Effectiveness of Training Strategies and Data** The **Qwen2.5-7B-Base-CPT** model is trained on our AnesCorpus, with **Qwen2.5-7B-Base** serving as the foundation model.

| Model | SFT Data | | AnesBench | |
|---|---|---|---|---|
| | AnesQA | Medical-o1 | English | Chinese |
| **Qwen2.5-7B-Base** | ✓ | ✗ | 49.3 | 64.9 |
| | ✗ | ✓ | 49.1 | 63.0 |
| | ✓ | ✓ | 49.7 | 65.9 |
| **Qwen2.5-7B-Base-CPT** | ✓ | ✗ | 49.7 | 50.7 |
| | ✗ | ✓ | 50.7 | 59.4 |
| | ✓ | ✓ | 51.2 | 60.0 |

**Complementarity of AnesQA and Medical-o1.** The results underscore the complementary nature of AnesQA and Medical-o1 for fine-tuning. As demonstrated, integrating AnesQA with Medical-o1 leads to varying degrees of performance improvement across models. This finding suggests that, although Medical-o1 is not exclusively focused on anesthesiology, its inclusion of general medical knowledge is advantageous for enhancing the model's anesthesiological capabilities, thereby complementing the specialized content provided by AnesQA. This synergy highlights that, even within highly specialized domains such as anesthesiology, the incorporation of broader medical knowledge remains essential.

**Impact of Continued Pre-training.** It was found that continued pre-training with AnesCorpus has a dichotomous effect on multilingual performance. While it enhances performance on the AnesBench-English, raising the maximum accuracy from 49.7% on the base model to 51.2% after CPT, it concurrently degrades performance on the AnesBench-Chinese. This demonstrates that domain-specific pre-training does not uniformly guarantee improvements across all languages. We hypothesize that this disparity stems from the model's inability to effectively transfer knowledge across languages; the gains in English proficiency are not successfully propagated to the Chinese counterpart, potentially inducing catastrophic forgetting. Consequently, we recommend that the linguistic distribution within CPT corpora be carefully managed to avoid compromising the model's multilingual versatility.

## 6 LIMITATIONS

While AnesSuite contributes to anesthesiology reasoning with LLMs, several limitations remain. First, the System 2 questions are constructed from abstract scenarios rather than real clinical cases. Second, future iterations of AnesSuite should consider incorporating additional modalities, to better reflect real-world clinical environments.

## 7 CONCLUSION

This paper introduces AnesSuite, the first comprehensive suite dedicated to anesthesiology reasoning. AnesSuite features a structured evaluation benchmark and three training datasets, establishing a complete infrastructure for key development stages: CPT, SFT, and RLVR. We also developed Morpheus, the first strong baseline model collection for this domain. Trained exclusively on the AnesR1, these models demonstrate consistent performance gains across anesthesiology, general medical, and general-domain benchmarks. Furthermore, extensive ablation studies and analyses yield several key insights for improving these specialized reasoning capabilities. We hope this work will serve as a cornerstone for developing LLMs with enhanced anesthesiology reasoning capabilities.

## ACKNOWLEDGEMENTS

This work was supported in part by the New Generation Artificial Intelligence-National Science and Technology Major Project (Grant No. 2025ZD0123602), the National Natural Science Foundation of China (Grant No. U23A20318 and 62276195), the Science and Technology Major Project of Hubei Province (Grant No. 2024BAB046 and 2025BCB026), the Foundation for Innovative Research Groups of Hubei Province (Grant No. 2024AFA017), the Medical Health Science and Technology Project of Zhejiang Province (Grant No.WKJ-ZJ-26004) and the Major Science and Technology Special Plan Project, Department of Science and Technology of Yunnan Province (Grant No. 202502AS080002). This work was also supported by WHU-Kingsoft Joint Lab. The numerical calculations in this paper have been done on the supercomputing system in the Supercomputing Center of Wuhan University.

We would like to thank Professor Bo Du for his helpful discussions throughout this work. Furthermore, we extend our gratitude to Tao Huang and Zhiwei Wang for their assistance in the data collection process.

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

# Appendix

## Contents

# A DATASETS CURATION DETAILS

## A.1 DATA LEAKAGE ANALYSIS

Given the increasing volume of pre-training data utilized by large language models, coupled with the fact that the sources and specific acquisition methods are often regarded as core secrets and rarely disclosed, a pertinent issue arises: whether the questions in our benchmark have been inadvertently leaked into the pre-training data of various models. We have employed a method specifically designed for multiple-choice questions to detect potential data leakage (Ni et al., 2025). For a given multiple-choice instance $x = (x_{\text{stem}}, o_1, o_2, \ldots, o_n)$, where $x_{\text{stem}}$ represents the question stem and $o_1, o_2, \ldots, o_n$ are the options, we generate all possible permutations of the options, resulting in $\{x_1, x_2, \ldots, x_{n!}\}$. For each permutation, the sequence is decomposed into tokens as $x = (t_1, t_2, \ldots, t_m)$. We then compute the confidence score for each sequence using the following method:

$$\text{Conf}_{\text{LLM}}(x) = \left( \prod_{i=1}^{m-1} p_{\text{LLM}}(t_i|x) \right)^{\frac{1}{m-1}}. \tag{1}$$

If the original order $x_0$ has the highest confidence score among all permutations, we consider that the multiple-choice instance **may** have experienced data leakage. We randomly select 500 questions from AnesBench, excluding those with excessive options. Under the assumption that the LLM has no prior knowledge and calculates probabilities randomly, this method yields a potential leakage proportion of 0.04. As shown in Fig. 8, most models exhibit a potential data leakage proportion around 0.10, with the exception of the Qwen and Yi model families, which are slightly higher (consistent with the evaluation results of MMLU(Wang et al., 2023a), CMMLU(Li et al., 2024), C-Eval(Huang et al., 2023), and CMB(Wang et al., 2024a) in the original text (Ni et al., 2025)). Overall, the low potential leakage proportions of most models on AnesBench validate the quality of our work.

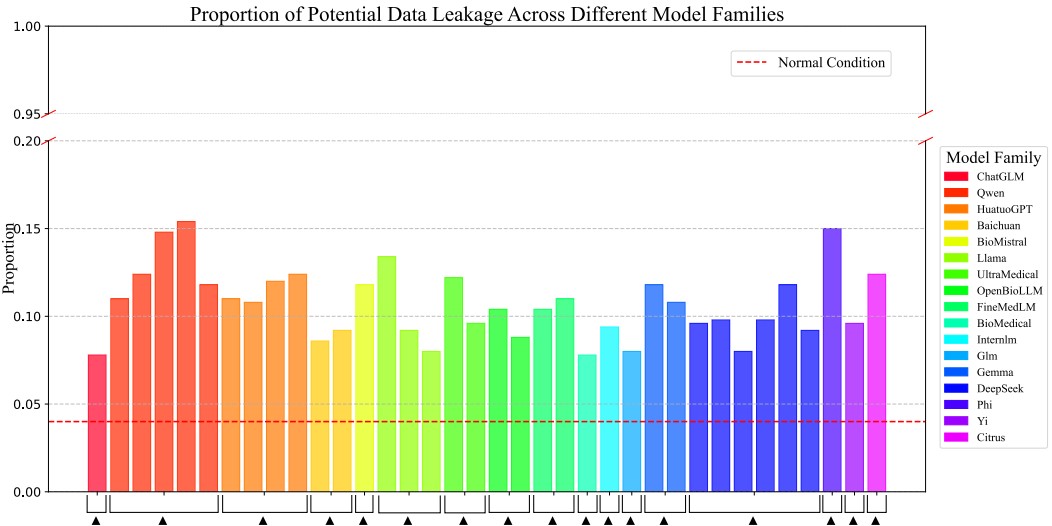

Figure 8: **Proportion of potential data leakage in the model**. Each ▲ below the x-axis denotes a model family.

## A.2 DETAILS OF ANESBENCH AND ANESR1 ANNOTATION

We annotated AnesBench and AnesR1 using detailed few-shot prompts with the inference model DeepSeek-R1. The specific prompts are as follows.

---

**Annotation Prompt**

**SYSTEM PROMPT**

```
You are a anesthesiologist and helpful personal assistant,
    ↪ proficient in categorizing questions strictly according to
    ↪ user instructions.
```

**USER PROMPT**

```
Problem description:
[Insert the question stem and all options here]

Answer:
[Insert the correct answer here]

Classification Categories:
1. Knowledge-Based: Involves recalling fundamental facts,
    ↪ definitions, or procedures. This type of question primarily
    ↪ tests factual memory without requiring application or in-
    ↪ depth reasoning. For example: "What is the standard dose of
    ↪ Propofol for an adult patient undergoing general anesthesia?"
2. Application-Based: Involves using basic knowledge or concepts in
    ↪  straightforward situations. The question requires applying
    ↪ known information to solve problems or make decisions, but
    ↪ does not require complex thinking or multiple-step reasoning.
    ↪  For example: "If a patient has a history of allergies to
    ↪ local anesthetics, which alternative drug should be
    ↪ considered?"
3. Reasoning-Based: Requires logical thinking, analysis, and the
    ↪ use of problem-solving skills. This type of question
    ↪ challenges the ability to evaluate complex situations, reason
    ↪  through different factors, and make informed decisions. For
    ↪ example: "Given the patient's medical history and lab results
    ↪ , determine the most appropriate anesthetic technique for the
    ↪  upcoming surgery."

Instructions for Classification:
Please categorize the problem by selecting the most appropriate
    ↪ category that aligns with the type of thinking and approach
    ↪ required to address the question. Consider factors such as
    ↪ the level of complexity, the type of cognitive processing
    ↪ needed, and whether the problem involves straightforward
    ↪ recall, application, or in-depth reasoning. Your response
    ↪ should conclude with: "So, the problem can be categorized as
    ↪ $\boxed{ANSWER}$," where ANSWER is one of the numbers 1, 2,
    ↪ or 3.
```

### A.3    DETAILS OF MANUAL INSPECTION IN ANESBENCH

**General Review**    To ensure the data quality of AnesBench, we selected approximately 60% of the questions for manual inspection. Specifically, the types of errors we identified and their proportions among the inspected questions are shown in Tab 6. The proportions of the above three types of errors are all very low, which can basically guarantee the data quality of AnesBench. All the problematic questions we have checked above has been excluded from AnesBench.

Table 6: Summary of Manual Inspection on AnesBench

| Issue Type | Count | Percentage |
|---|---|---|
| Format Error | 1 | 0.02% |
| Content Mistake | 5 | 0.1% |
| Missing Key Information | 11 | 0.22% |

## A.4 DETAILS OF ANESCORPUS CURATION

To construct AnesCorpus, we first retrieved anesthesia-related documents from FineWeb and Chinese FineWeb based on manually selected prompts. Then, through a two-stage data cleaning process on AnesBench, we finally obtained AnesCorpus. The two sets of keywords we used in the first stage are as follows. The first set of keywords is highly relevant to anesthesiology. The second set is closely related to perioperative management in anesthesiology, which helps us not only focus on theoretical knowledge of anesthesiology but also obtain more content related to clinical perioperative management.

| Group | Keyword |
|---|---|
| Keywords1 | 麻醉,神经阻滞,镇静 ,镇痛 ,anesthe,analge ,sedation ,nerve block |
| Keywords2 | 手术 , 围术期 ,surgery,surgical,operation, operative |

Table 7: Keywords used for AnesCorpus Construction

**Decontamination protocol** We employed a two-stage decontamination protocol to eliminate potential data contamination. The first stage consisted of a preliminary screening based on n-gram overlap. Specifically, we constructed an inverted index of n-grams from the AnesBench evaluation dataset, mapping each n-gram fragment to the set of questions in which it appears. We then processed the initial corpus by querying each document's n-grams against this index to compute the number of shared n-gram fragments with AnesBench. Documents exceeding a predefined threshold were flagged as potentially contaminated.

The second stage implemented a more stringent filtering procedure, following the methodology of HuatuoGPT-o1 (Chen et al., 2024). In this phase, any document exhibiting a LCS greater than 64 characters with any item in AnesBench was definitively excluded from the corpus. In accordance with this protocol, we set the character-level n-gram size to 35 and the overlap threshold to 9. During the first stage, we identified 359,598 suspicious candidates. In the second stage, we further screened and detected a total of 3,240 contaminated documents that exhibited an overlap with AnesBench greater than 64 characters.

## A.5 DETAILS OF ANESQA CURATION

AnesQA was constructed using a multi-stage, LLM-based pipeline to transform existing anesthesiology literature into question-answer pairs. The generation process began by segmenting source documents into smaller chunks based on length and section structure. For each chunk, LLaMA3.3-70B-Instruct (Grattafiori et al., 2024) was prompted to generate self-contained questions. Subsequently, a distinct model, Qwen2.5-72B-Instruct (Yang et al., 2024), was employed to filter these questions, selecting only the most informative and domain-relevant ones to answer. This two-model approach was implemented to mitigate potential self-preference bias.

To address the risk of model-induced hallucinations, we instituted several quality assurance protocols, including model cross-validation and manual quality assessments on a sampled subset. The value of AnesQA is further substantiated by its demonstrated ability to enhance the reasoning performance of LLMs, as detailed in the main text. This finding serves as an indirect validation of the dataset's quality and its utility for domain adaptation. The specific prompts used for the generation and answering stages are provided below.

---

**AnesQA Generation Prompt**

**SYSTEM PROMPT**

```
Based on the following text, generate a series of closely related
    ↪ questions. Ensure that the questions are phrased in the same
    ↪ language as the text and cover the following types:

Knowledge-based questions: Assessing fundamental knowledge and key
    ↪ concepts.
```

```
Scenario-based questions: Designing practical application scenarios
    ↪  based on the text content.

Decision-making questions: Proposing questions that require
    ↪ judgment or selection under specific conditions.

Inference-based questions: Asking for logical reasoning based on
    ↪ the text information.

Open-ended questions: Encouraging in-depth thinking and discussion.
Ensure that the questions are varied in nature, particularly the
    ↪ scenario-based questions, which should reflect a broad range
    ↪ of potential real-world situations related to the text's
    ↪ context.

The diversity of the questions should cover different angles and
    ↪ contexts, enhancing the complexity and richness.
Additionally, retain the full names of abbreviations in the
    ↪ questions.
Ensure that the questions are closely related to anesthesia.

Example Output:
Knowledge-based questions: What are the key phases of general
    ↪ anesthesia, and what is their purpose?

Scenario-based questions: Imagine a pediatric patient undergoing
    ↪ surgery. How would the anesthetic approach differ from that
    ↪ of an elderly patient?

Decision-making questions: If a patient has a history of adverse
    ↪ reactions to anesthesia, what considerations should guide the
    ↪  selection of anesthetic agents?

Inference-based questions: Why might patient-specific factors such
    ↪ as weight and medical history significantly influence
    ↪ anesthesia management?
Open-ended questions: What are the potential challenges in
    ↪ balancing safety and efficacy in anesthesia for high-risk
    ↪ patients?
```

**USER PROMPT**

```
The provided text is as follows:

[Insert the text here for generating related questions]
```

---

**AnesQA Filtering and answering Prompt**

**SYSTEM PROMPT**

```
Based on the set of questions generated from the given text, select
    ↪  3 to 5 question that the most related to anesthesia and
    ↪ provide a clear, well-structured answer for it.
Ensure that each question and answer is highly relevant to the
    ↪ field of anesthesia. The responses should address the core
    ↪ principles, practices, and challenges in anesthesia, and
    ↪ should not diverge into unrelated topics.
Ensure the following:
Knowledge-based questions: Choose a question that asks for a
    ↪ fundamental concept or core knowledge and provide a precise,
    ↪ informative answer that fully explains the concept without
    ↪ referring back to the original text.
```

```
Scenario-based questions: Select a question that involves a
    ↪ practical, real-world application and provide a thoughtful
    ↪ response, addressing all aspects of the scenario described
    ↪ and providing a complete solution or explanation.
Decision-making questions: Pick a question that requires judgment
    ↪ or decision-making and offer a reasoned, evidence-based
    ↪ answer. Ensure your response fully justifies the choice and
    ↪ addresses the relevant factors involved in making the
    ↪ decision.
Inference-based questions: Choose a question that requires logical
    ↪ reasoning or drawing conclusions, and provide a well-
    ↪ explained answer that walks through the reasoning process and
    ↪  clearly supports the conclusion, without relying on
    ↪ information outside the answer itself.
Open-ended questions: Pick a question that encourages in-depth
    ↪ thinking and discussion, and provide a comprehensive answer
    ↪ that explores multiple perspectives and offers a complete
    ↪ exploration of the topic, fully addressing all aspects of the
    ↪  question.
Each answer should be complete and self-contained, without indirect
    ↪  references to the original text. For each question,
    ↪ explicitly state which question has been selected, and then
    ↪ provide a detailed, comprehensive answer.
For each question, follow this structure:
**Selected Question:** [State the exact question chosen]
**Selected Question Type:** [State the question type, e.g.,
    ↪ Knowledge-based, Scenario-based, Decision-making, Inference-
    ↪ based, Open-ended]
**Answer:** [Provide the complete answer without referring back to
    ↪ the original text.]
```

**USER PROMPT**

```
The provided set of questions is as follows:
[Insert the set of generated questions here]
```

To ensure data quality and integrity, we performed a comprehensive technical review of the AnesQA dataset. Initially, a manual inspection of 100 randomly sampled QA pairs revealed a specific formatting error. Based on this finding, we developed a regular expression to systematically scan the entire dataset for all instances of this and similar errors. This automated process identified 119 problematic QA pairs, which were then expunged from the dataset. The resulting refined AnesQA dataset, consisting of over 20,000 QA pairs, is well-suited for its designated purpose.

| Model Group | Models |
|---|---|
| Qwen | Qwen3-0.6B, Qwen3-1.7B, Qwen3-4B, Qwen3-8B, Qwen3-14B, Qwen3-30B-A3B, Qwen3-32B, Qwen3-235B-A22B (Team, 2025),QwQ-32B-Preview (Qwen, 2025) Qwen2.5-7B-Instruct, Qwen2.5-14B-Instruct, Qwen2.5-32B-Instruct, Qwen2.5-72B-Instruct (Yang et al., 2024) |
| DeepSeek | DeepSeek-R1-Distill-Qwen-1.5B, DeepSeek-R1-Distill-Qwen-7B, DeepSeek-R1-Distill-Qwen-14B, DeepSeek-R1-Distill-Qwen-32B, DeepSeek-R1-Distill-Llama-70B, Deepseek-R1 (Guo et al., 2025), DeepsSeek-V3 (Liu et al., 2024a) |
| Gemma | Gemma-3-1b-it, Gemma-3-4b-it, Gemma-3-12b-it, Gemma-3-27b-it (Team et al., 2025a) Gemma-2-9b-it, Gemma-2-27b-it (Team et al., 2024) |
| Llama | Meta-Llama-3-8B-Instruct, Llama-3.1-8B-Instruct, Llama-3.3-70B-Instruct (Grattafiori et al., 2024), Llama-4-Scout-17B-16E ,Llama-4-Maverick-17B-128E (Meta, 2025) |
| HuatuoGPT | HuatuoGPT-o1-7B, HuatuoGPT-o1-8B, HuatuoGPT-o1-70B, HuatuoGPT-o1-72B (Chen et al., 2024) |
| Other Chinese Models | chatglm3-6b, glm-4-9b-chat (GLM et al., 2024), internlm3-8b-instruct (Cai et al., 2024), Baichuan2-7B-Chat, Baichuan2-13B-Chat (Baichuan, 2023), Yi-1.5-34B-Chat (Young et al., 2024) |
| Other Medical Models | BioMistral-7B (Labrak et al., 2024), FineMedLM, FineMedLM-o1 (Yu et al., 2025), Citrus1.0-llama-70B (Wang et al., 2025), Llama3-OpenBioLLM-8B, Llama3-OpenBioLLM-70B (Ankit Pal, 2024), Llama-3-70B-UltraMedical, Llama-3.1-8B-UltraMedical (Zhang et al., 2024), Bio-Medical-Llama-3-8B (ContactDoctor, 2024) |
| Proprietary Models | GPT-4o (OpenAI, 2024a), Gemini-2.5-Flash, Gemini-2.5-Pro (Co-manici et al., 2025), Claude-3.7-Sonnet (Anthropic, 2025) |
| Other General Models | Phi-4 (Abdin et al., 2024) |

Table 8: List of Evaluation Models

# B  EVALUATION DETAILS

## B.1  LIST OF EVALUATION MODELS

In addition to the results in Tab 3 in the main text, we have evaluated the anesthesiology reasoning performance of more than 50 LLMs using AnesBench-English in total. The complete list of evaluated models is shown in Tab. 8.

## B.2  EXPERIMENTAL ENVIRONMENT FOR EVALUATION

The evaluation experiments were primarily conducted on a server equipped with eight NVIDIA A800 GPUs. The sglang framework (Zheng et al., 2024a) was employed for inference. To ensure deterministic and reproducible outputs, the sampling temperature was set to 0, with the maximum number of output tokens limited to 2048. We adopted the Zero-Shot COT methodology for prompting (Wei et al., 2022). For proprietary and larger models (e.g., Llama-4-Maverick-17B-128E), inference was performed via their respective APIs.

## B.3  COMPLETE EVALUATION RESULTS

The complete evaluation results on AnesBench-English are shown in Tab. 9.

Table 9: **Evaluation Results on AnesBench-English**

| Model | Sys1 | Sys1.x | Sys2 | Total |
|---|---|---|---|---|
| Qwen3-0.6B | 0.39 | 0.31 | 0.26 | 0.36 |
| Qwen3-1.7B | 0.48 | 0.37 | 0.30 | 0.44 |
| Qwen3-4B | 0.60 | 0.46 | 0.34 | 0.54 |
| Qwen2.5-7B-Instruct | 0.56 | 0.44 | 0.36 | 0.52 |
| Qwen3-8B | 0.65 | 0.50 | 0.40 | 0.60 |
| Qwen2.5-14B-Instruct | 0.61 | 0.52 | 0.41 | 0.57 |
| Qwen3-14B | 0.70 | 0.57 | 0.45 | 0.65 |
| Qwen3-30B-A3B | 0.73 | 0.60 | 0.48 | 0.68 |
| Qwen3-32B | 0.72 | 0.64 | 0.48 | 0.68 |
| QwQ-32B-Preview | 0.69 | 0.58 | 0.44 | 0.64 |
| Qwen2.5-32B-Instruct | 0.65 | 0.55 | 0.44 | 0.61 |
| Qwen2.5-72B-Instruct | 0.72 | 0.60 | 0.48 | 0.67 |
| Qwen3-235B-A22B | 0.78 | 0.67 | 0.57 | 0.74 |
| DeepSeek-R1-Distill-Qwen-1.5B | 0.32 | 0.28 | 0.27 | 0.31 |
| DeepSeek-R1-Distill-Qwen-7B | 0.40 | 0.34 | 0.28 | 0.38 |
| DeepSeek-R1-Distill-Qwen-14B | 0.64 | 0.51 | 0.40 | 0.59 |
| DeepSeek-R1-Distill-Qwen-32B | 0.67 | 0.56 | 0.45 | 0.63 |
| DeepSeek-R1-Distill-Llama-70B | 0.77 | 0.68 | 0.56 | 0.73 |
| DeepSeek-V3 | 0.77 | 0.69 | 0.55 | 0.73 |
| DeepSeek-R1 | 0.85 | 0.78 | 0.70 | 0.82 |
| Gemma-3-1b-it | 0.33 | 0.26 | 0.21 | 0.30 |
| Gemma-3-4b-it | 0.46 | 0.36 | 0.35 | 0.42 |
| Gemma-2-9b-it | 0.53 | 0.40 | 0.36 | 0.49 |
| Gemma-3-12b-it | 0.56 | 0.46 | 0.36 | 0.52 |
| Gemma-2-27b-it | 0.60 | 0.43 | 0.36 | 0.54 |
| Gemma-3-27b-it | 0.63 | 0.52 | 0.40 | 0.58 |
| Meta-Llama-3-8B-Instruct | 0.54 | 0.42 | 0.39 | 0.50 |
| Llama-3.1-8B-Instruct | 0.58 | 0.45 | 0.36 | 0.53 |
| Llama-3.1-8B-UltraMedical | 0.63 | 0.47 | 0.41 | 0.57 |
| Llama-3.3-70B-Instruct | 0.74 | 0.63 | 0.51 | 0.70 |
| Llama-3-70B-UltraMedical | 0.73 | 0.60 | 0.47 | 0.68 |
| Llama3-OpenBioLLM-70B | 0.68 | 0.55 | 0.44 | 0.63 |
| Llama-4-Scout-17B-16E | 0.77 | 0.66 | 0.55 | 0.72 |
| Llama-4-Maverick-17B-128E | 0.83 | 0.73 | 0.64 | 0.79 |
| HuatuoGPT-o1-7B | 0.56 | 0.45 | 0.38 | 0.52 |
| HuatuoGPT-o1-8B | 0.58 | 0.46 | 0.39 | 0.53 |
| HuatuoGPT-o1-70B | 0.70 | 0.58 | 0.48 | 0.65 |
| HuatuoGPT-o1-72B | 0.71 | 0.61 | 0.48 | 0.67 |
| chatglm3-6b | 0.37 | 0.28 | 0.25 | 0.34 |
| glm-4-9b-chat | 0.48 | 0.36 | 0.36 | 0.44 |
| internlm3-8b-instruct | 0.60 | 0.43 | 0.40 | 0.54 |
| Baichuan2-7B-Chat | 0.39 | 0.31 | 0.30 | 0.37 |
| Baichuan2-13B-Chat | 0.42 | 0.31 | 0.34 | 0.39 |
| Yi-1.5-34B-Chat | 0.54 | 0.44 | 0.35 | 0.50 |
| BioMistral-7B | 0.43 | 0.30 | 0.32 | 0.39 |
| FineMedLM | 0.40 | 0.35 | 0.27 | 0.38 |
| FineMedLM-o1 | 0.43 | 0.34 | 0.26 | 0.39 |
| Citrus1.0-llama-70B | 0.71 | 0.60 | 0.52 | 0.67 |
| Llama3-OpenBioLLM-8B | 0.44 | 0.35 | 0.30 | 0.41 |
| Bio-Medical-Llama-3-8B | 0.53 | 0.41 | 0.38 | 0.49 |
| GPT-4o | 0.81 | 0.72 | 0.59 | 0.77 |
| Gemini-2.5-Flash | 0.84 | 0.76 | 0.68 | 0.81 |
| Gemini-2.5-Pro | 0.89 | 0.82 | 0.77 | 0.86 |
| Claude-3.7-Sonnet | 0.80 | 0.73 | 0.63 | 0.77 |
| phi-4 | 0.69 | 0.57 | 0.41 | 0.64 |

## C TRAINING EXPERIMENT DETAILS

### C.1 EXPERIMENT DETAILS OF USING ANESQA FOR SFT

In the ablation study presented in Section 5.2, we utilized the AnesQA dataset as domain-specific data to evaluate its effectiveness in enhancing the anesthesiology reasoning capabilities of LLMs. As AnesQA consists exclusively of English question-answer pairs, we translated it into Chinese using GPT-4o-mini (OpenAI, 2024b). This step was taken to ensure the reliability of our results on the AnesBench-Chinese by mitigating the influence of language distribution and to align with the bilingual configuration of the Medical-o1 dataset. The specific prompt used for this translation process is provided below.

> **Prompt used for Translating AnesQA from English to Chinese**
>
> **SYSTEM PROMPT**
>
> ```
> You are a professional translator with expertise in medical and
>     ↪ anesthesia-related terminology.
> Your task is to accurately translate anesthesia-related QA pairs
>     ↪ from English to Chinese,
> ensuring that professional terms are translated precisely and
>     ↪ consistently.
> Maintain the original meaning, tone, and level of technical detail
>     ↪ while ensuring readability for a Chinese-speaking medical
>     ↪ audience.
> Do not add or omit any information.
> ```
>
> **USER PROMPT**
>
> ```
> Please translate the following anesthesia-related QA pair from
>     ↪ English to Chinese,
> ensuring that all medical terms, abbreviations, and technical
>     ↪ expressions are translated precisely and appropriately for a
>     ↪ professional audience.
> Maintain clarity, coherence, and fidelity to the original meaning.
>
> Input (English QA Pair):
>
> [Insert the English QA Pair here]
> ```

### C.2 EXPERIMENT DETAILS FOR BASELINE MODEL

The training of the baseline model comprises two primary stages: SFT and GRPO. All training experiments were conducted on a server with 8 **NVIDIA A800 (80G)** GPUs. For the SFT stage, we employed the Llamafactory framework (Zheng et al., 2024b). The key training hyperparameters are detailed in Tab. 10.

Table 10: Key Training Hyperparameters of SFT

| Category | Hyperparameter | Value |
|---|---|---|
| **Training Setup** | Cutoff Length | 4096 |
| | Num Train Steps | 100 |
| **Optimization** | Learning Rate | $1.0 \times 10^{-5}$ |
| | Warmup Ratio | 0.175 |
| | LR Scheduler | cosine |
| **Batch** | Per-device Train Batch Size | 16 |
| | Gradient Accumulation Steps | 4 |

In the second stage, we employed the Verl framework (Sheng et al., 2025) for model training using GRPO. The primary training parameters were configured as follows: a training batch size of 512 and

a mini-batch size of 256 for each gradient update. The maximum response length was constrained to 2048 tokens. For each prompt, the group size for rollout was set to 5. The training process was conducted for a maximum of 4 epochs, with a learning rate of $1 \times 10^{-6}$ and a learning rate warmup ratio of 0.3. Under this configuration, the Morpheus-7B, Morpheus-32B and Morpheus-14B models were trained for 50, 50 and 40 steps, respectively. For the Morpheus-32B, due to the constraints of computing resources, we adopted the parameter-efficient fine-tuning method of LoRA. Among them, the rank $r$ of matrix factorization was set to 128, and the scaling factor $\alpha$ was set to 32.

In our specific training process, we exclusively employed an outcome-based binary reward mechanism: a reward of 1 is assigned if and only if the model's final output matches the ground truth; otherwise, the reward is 0. To accurately extract the final answer from the model's output, we utilized robust regular expressions to strictly match patterns such as "Answer is" or "Answer:". To demonstrate the stability of the training process, we conducted three independent training runs for the Morpheus-7B model. As illustrated by the reward curves in Fig 9 and Fig 10, the training process exhibits significant stability across both the 7B and 14B models.

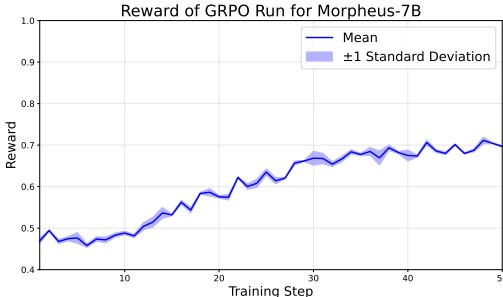
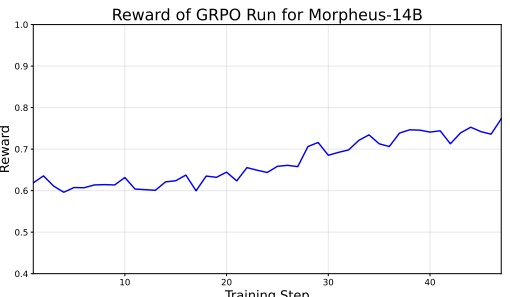

Figure 9: Reward curve of Morpheus7B      Figure 10: Reward curve of Morpheus14B

Furthermore, we trained multiple models using the AnesR1 dataset with the number of training steps extended to 200. As shown in Tab 11, the results demonstrate significant performance improvements across both the Chinese and English subsets of AnesBench, validating the high quality and rich knowledge encapsulated within AnesR1. Notably, as shown in Tab. 4, the Qwen model exhibited a decline in the Chinese subset following SFT, which may be attributed to an alignment tax resulting from an insufficient number of SFT steps intended for domain alignment.

| Model | AnesBench-English | AnesBench-Chinese | Avg. |
|---|---|---|---|
| Llama-3.1-8B-Instruct | 0.54 | 0.48 | 0.51 |
| Llama-3.1-8B(AnesR1) | 0.60 | 0.51 | 0.55 |
| Gemma2-2B-Instruct | 0.40 | 0.34 | 0.37 |
| Gemma2-2B(AnesR1) | 0.45 | 0.41 | 0.43 |
| Gemma3-4B-Instruct | 0.32 | 0.40 | 0.36 |
| Gemma3-4B(AnesR1) | 0.36 | 0.43 | 0.39 |

Table 11: Additional Training Experiments with AnesR1

Finally, we conducted supplementary experiments using the InternLM3-8B-Instruct (Cai et al., 2024) model. The training dataset employed was AnesR1, consistent with the setup used to train the Morpheus model in the main text. The number of training steps for SFT and GRPO was set to 100 and 60, respectively. As illustrated in Tab. 12, following SFT and GRPO training on AnesR1, the model demonstrated significant improvements across all three cognitive dimensions, thereby indicating the strong adaptability of our dataset across diverse foundation models.

| Model | System1 | System1.x | System2 | Overall |
|---|---|---|---|---|
| InternLM3-8B-Instruct | 0.59 | 0.45 | 0.39 | 0.54 |
| InternLM3-8B-Instruct (SFT + RL) | 0.62 | 0.50 | 0.43 | 0.58 |

Table 12: Results of fine-tuned InternLM3 on AnesBench-English

## C.3 EXPERIMENT DETAILS FOR ABLATION STUDY

**Hardware and Software** All experiments were conducted on 8 **NVIDIA A800 (80G)** GPUs, interconnected via NVLink. We utilized Llama-Factory as the training infrastructure. Performance was further enhanced by leveraging BF16 mixed-precision, FlashAttention-2, and the LIGER kernel.

**Continued Pre-training (CPT)** The CPT stage involved full-parameter fine-tuning of the **Qwen2.5-7B** base model. We set the maximum sequence length to 2048 tokens. The model was trained for 3 epochs using DeepSpeed with a ZeRO Stage 1 configuration. Key hyperparameters included an effective batch size of 64 (a per-device batch size of 8 with 8 gradient accumulation steps), a peak learning rate of $1.0 \times 10^{-4}$ decayed with a cosine annealing schedule, and a linear warmup phase over the initial 5% of training steps.

**Supervised Fine-tuning (SFT)** Starting from the CPT model, we performed full-parameter supervised fine-tuning. The maximum sequence length was increased to 4096 tokens. This stage ran for 5 epochs, employing a more memory-efficient DeepSpeed ZeRO Stage 3 configuration. The hyperparameters were adjusted to an effective batch size of 32 (a per-device batch size of 8 with 4 accumulation steps) and a reduced peak learning rate of $1.0 \times 10^{-5}$, also with a cosine schedule. The warmup period was extended to 10% of the total steps, and gradient clipping with a maximum norm of 5.0 was applied to ensure stability.

## D ANESCORPUS VISUALIZATION

As illustrated in Fig. 11, our constructed anesthesia-related corpus encompasses a diverse range of medical terminologies, both in English and Chinese. The English word cloud highlights key terms such as "general anesthesia", "local anesthetic", "surgical procedure", and "pain management", reflecting a strong focus on clinical applications, patient care, and perioperative considerations. Additionally, frequent mentions of terms like "nerve block", "blood pressure", and "side effect" indicate a broad coverage of anesthesia techniques and associated physiological impacts. Similarly, the Chinese word cloud presents a comparable distribution, with dominant terms such as "手术(surgery)", "治疗(treatment)", "麻醉(anesthesia)", indicating a strong alignment with procedural and treatment-related aspects. The presence of terms related to medical interventions, such as "气管插管(intubation)" and "药物治疗(medication)", exhibits emphasis on clinical procedures and pharmacological considerations.

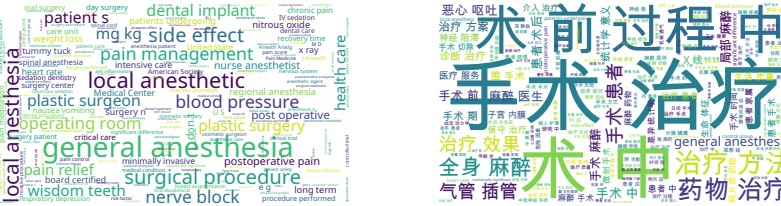

Figure 11: Word clouds of CPT corpus.

## E CASE STUDY

To highlight the challenges inherent in specialized medical reasoning, we conducted a comparative case study featuring Gemini-2.5-Pro, Qwen2.5-14B-Instruct and Morpheus-14B. The cases

presented are those in which Morpheus-14B provided the correct answer while other two models did not. The objective of this analysis is not to assert the overall superiority of Morpheus-14B. Rather, it is to illustrate that for a highly specialized discipline such as anesthesiology, professional knowledge and general reasoning capabilities are insufficient. Subtle distinctions in clinical scenarios can lead to fundamentally different outcomes, underscoring the need for more advanced and specialized reasoning.

For example, in Case 1, Gemini-2.5-Pro demonstrates clear reasoning and accurately recognizes that the $\mu$-opioid receptor is blocked. However, it fails to acknowledge that naltrexone exhibits affinity not only for the $\mu$ receptor but also for the $\kappa$ and $\delta$ opioid receptors. As a result, the combined use of naltrexone and nalbuphine cannot achieve an analgesic effect. Consequently, option C-"Continue naltrexone with small doses of nalbuphine every 4 hours as needed", is incorrect.

In Case 2, the reasoning of Gemini-2.5-Pro was almost correct; however, it failed to recognize that its rationale is valid only in the context of severe hypotension. According to the case description, the clinician's primary concern should be pump failure and hypoperfusion rather than hypotension. As a result, the model's recommendation was premature in the specific clinical scenario.

The following are the specific contents of the two cases.

---

**Case 1: Postoperative Pain Management**

**Question:** A 37-year-old man is brought to the operating room for repair of a broken mandible sustained in a motor vehicle accident. No other injuries are significant. The patient has been in treatment for alcohol abuse and takes disulfiram and naltrexone. Which of the following would be the best technique for management of this patient's postoperative pain?
**A.** Continue naltrexone with round-the-clock low-dose methadone **B.** Continue naltrexone with small doses of morphine every 4 hours as needed **C.** Continue naltrexone with small doses of nalbuphine every 4 hours as needed **D.** Discontinue naltrexone and treat pain with morphine as needed
**Correct Answer:** D

---

**Gemini-2.5-Pro (Incorrect):** This is a multiple-choice question about postoperative pain management in a patient on naltrexone. Let's break down the options step-by-step.
**Patient Profile:** * 37-year-old male undergoing repair of a broken mandible. This is expected to be a painful procedure. * History of alcohol abuse. * Current medications: Disulfiram and Naltrexone.
**Key Pharmacological Concepts:** * **Naltrexone:** A competitive opioid antagonist, primarily at the mu-opioid receptor. It is used to prevent relapse in alcohol and opioid use disorders. By blocking mu-receptors, it will render typical opioid agonists (like morphine, fentanyl, oxycodone) ineffective at standard doses. * **Morphine & Methadone:** These are pure mu-opioid receptor agonists. Their analgesic effects are mediated through the mu-receptor. * **Nalbuphine:** A mixed agonist-antagonist opioid. It is an agonist at the kappa-opioid receptor (which provides analgesia) and an antagonist at the mu-opioid receptor.
**Analysis of Options:**
* **A. Continue naltrexone with round-the-clock low-dose methadone:** This is incorrect. Methadone is a mu-opioid agonist. Naltrexone is a mu-opioid antagonist. Administering methadone to a patient on naltrexone will result in a lack of analgesic effect because the mu-receptors are blocked by naltrexone.
* **B. Continue naltrexone with small doses of morphine every 4 hours as needed:** This is incorrect for the same reason as option A. Morphine is a pure mu-opioid agonist. Its analgesic effect will be blocked by naltrexone. Overcoming this blockade would require dangerously high doses of morphine, which is not a safe or predictable approach.
* **C. Continue naltrexone with small doses of nalbuphine every 4 hours as needed:** This is the most suitable option. Naltrexone primarily blocks the mu-opioid receptors. Nalbuphine provides analgesia through its agonist activity at the kappa-opioid receptors. Since naltrexone does not significantly block kappa receptors, the analgesic effect of nalbuphine should be preserved. This allows for effective pain management while continuing the naltrexone therapy, which is important for preventing alcohol relapse.
* **D. Discontinue naltrexone and treat pain with morphine as needed:** This is not the best approach. Naltrexone has a long duration of action (its active metabolite has a half-life of 13 hours). To allow morphine to be effective, naltrexone would need to be discontinued for at least 72 hours prior to surgery. This is often not feasible in an acute setting like a trauma case. Furthermore, discontinuing naltrexone, especially during a stressful period like surgery, significantly increases the patient's risk of relapsing to alcohol use. The goal should be to manage pain while maintaining the patient's substance abuse treatment if possible.

**Conclusion:** The best strategy among the choices is to use an analgesic that bypasses the mu-receptor blockade. Nalbuphine, with its kappa-agonist activity, fits this requirement. This approach allows for pain control without interrupting the patient's crucial naltrexone therapy for alcohol abuse.
**Final Answer:** The final answer is **(C)**

**Morpheus-14B (Correct):** First, the patient is a 37-year-old man with a broken mandible from a motor vehicle accident. He's in treatment for alcohol abuse and is taking disulfiram and naltrexone. I need to manage his postoperative pain.

Disulfiram is used to treat alcoholism by causing an unpleasant reaction when alcohol is consumed, so he shouldn't drink. Naltrexone is an opioid antagonist that blocks the effects of opioids, used to prevent relapse by reducing the rewarding effects of alcohol and opioids.

The key point is that naltrexone blocks opioid receptors, so if I give him opioids for pain, they won't work because naltrexone is antagonizing the receptors. That means I can't use standard opioid analgesics like morphine.

Now, looking at the options:

A. Continue naltrexone with round-the-clock low-dose methadone

B. Continue naltrexone with small doses of morphine every 4 hours as needed

C. Continue naltrexone with small doses of nalbuphine every 4 hours as needed

D. Discontinue naltrexone and treat pain with morphine as needed

Option B says to continue naltrexone and use morphine. But since naltrexone blocks opioid receptors, morphine won't be effective. That seems counterproductive.

Option C mentions nalbuphine. Nalbuphine is an opioid agonist-antagonist. It has partial agonist activity at kappa receptors and antagonist activity at mu receptors. Since naltrexone is already blocking mu receptors, nalbuphine might not work well because it's an antagonist at mu receptors too. Nalbuphine could actually precipitate withdrawal in opioid-dependent patients, but here it's for alcohol abuse, not opioid dependence. Still, with naltrexone blocking, it might not be effective.

Option A has methadone. Methadone is a full opioid agonist, but it's long-acting. If naltrexone is blocking, methadone might not work. Also, methadone is used for maintenance in opioid addiction, but here it's for pain, and with naltrexone, it could be ineffective.

Option D says to discontinue naltrexone and treat with morphine. That makes sense because if I stop naltrexone, the opioid receptors are free, and morphine can work. But I need to consider the context. He's in treatment for alcohol abuse, and naltrexone is part of that. Discontinuing it might compromise his treatment for alcoholism.

The question is about postoperative pain management. The surgery is for a broken mandible, which can be painful, but it's not life-threatening. I need to balance pain control with the patient's ongoing treatment.

Disulfiram is also mentioned, but it doesn't directly affect opioid management; it's for alcohol.

Naltrexone is the key here. It's an opioid antagonist, so any opioid I give will be blocked.

Therefore, I can't use opioids while he's on naltrexone. I need an alternative pain management strategy.

Option C has nalbuphine. Nalbuphine is an opioid agonist-antagonist. It has kappa agonist and mu antagonist properties. Since naltrexone is blocking mu receptors, nalbuphine might still have some kappa agonist effects, but it could also precipitate withdrawal or not be effective.

In practice, for patients on naltrexone, we avoid opioids and use non-opioid analgesics.

But let's see the options again. Option C says "small doses of nalbuphine every 4 hours as needed." Nalbuphine is sometimes used in such cases, but it's not ideal because of the antagonist properties. Perhaps I should look for a non-opioid option, but it's not listed.

Option D: discontinue naltrexone and use morphine. That seems straightforward, but it might not be ethical or practical to discontinue naltrexone just for pain management.

The patient has no other significant injuries, so pain is manageable.

I recall that in anesthesia, for patients on naltrexone, we often discontinue it before surgery and restart after, or use non-opioid analgesics.

But here, the options don't include non-opioids. They all involve some form of opioid or naltrexone continuation.

Option A has methadone, which is an opioid, so it won't work with naltrexone.

Option B has morphine, which won't work.

Option C has nalbuphine, which might have some efficacy but is not reliable.

Option D: discontinue naltrexone and use morphine. This might be the best choice because it allows effective pain control.

But discontinuing naltrexone could be problematic. Naltrexone is used to prevent relapse, so stopping it might increase the risk of alcohol relapse.

The surgery is for a broken mandible, which might not directly involve alcohol, but pain could be a stressor.

Perhaps the best approach is to discontinue naltrexone temporarily for the duration of the pain, and then restart it.

In many guidelines, for patients on naltrexone, it's recommended to hold the medication before elective surgery and restart after the acute pain period.

Since this is a motor vehicle accident, it might not be elective, but the question is about postoperative management.

The question says "repair of a broken mandible," so it's likely planned surgery, but the injury was from an accident.

I think the most suitable answer is D, discontinue naltrexone and treat with morphine as needed.

But let's see why the others are wrong.

A: Methadone – won't work with naltrexone.

B: Morphine – won't work.

C: Nalbuphine – might not work well due to antagonist properties, and could cause issues.

D: Discontinue naltrexone and use morphine – allows effective pain control.

Also, disulfiram is not relevant here.

So, I think D is the best choice.

But is it ethical to discontinue naltrexone? The question doesn't specify, so I have to go with the most effective pain management.

Perhaps in some contexts, we use non-opioid analgesics, but they're not options.

Option C has nalbuphine, which is an opioid, but it's not pure agonist.

Nalbuphine is sometimes used in patients on naltrexone because it has kappa agonism, but it can cause dysphoria or other issues.

I think D is the standard approach.

Final decision: D.

The patient is on naltrexone, an opioid antagonist used for alcohol abuse treatment, which blocks opioid receptors. This means standard opioid analgesics like morphine (options B and D) or methadone (option A) will be ineffective due to receptor blockade. Nalbuphine (option C) is an opioid agonist-antagonist with partial kappa agonist and mu antagonist properties; however, in the presence of naltrexone, it may not provide reliable analgesia and could precipitate withdrawal or cause dysphoria.

Option D involves discontinuing naltrexone temporarily for the postoperative period, allowing effective morphine use for pain control, and then restarting naltrexone after the acute pain resolves. This approach balances pain management with the patient's ongoing alcohol treatment, as the mandible repair is not life-threatening and pain can be managed with opioids without compromising long-term naltrexone therapy.

Thus, the best technique is to discontinue naltrexone and treat pain with morphine as needed.

**Answer: D**

**Qwen-2.5-14B-Instruct (Incorrect):** To determine the best technique for managing postoperative pain in this patient, we need to consider the pharmacological interactions and the patient's medical history.

1. **Disulfiram and Naltrexone**: The patient is taking both disulfiram and naltrexone for alcohol abuse. Disulfiram is an alcohol deterrent that causes unpleasant reactions if alcohol is consumed. Naltrexone is an opioid antagonist used to reduce cravings for alcohol and opioids.

2. **Opioid Management**: Since naltrexone is an opioid antagonist, it blocks the effects of opioids. Therefore, using traditional opioids like morphine would be less effective because naltrexone would reduce their analgesic effects.

3. **Alternative Analgesics**: Given that naltrexone is being used, we need to consider alternative analgesics that are not opioids or that have a different mechanism of action. Nalbuphine is a kappa-opioid receptor agonist and mu-opioid receptor antagonist, which means it can provide pain relief without being fully blocked by naltrexone.

Let's analyze each option:

- **Option A**: Continuing naltrexone with low-dose methadone. Methadone is an opioid and would be significantly reduced in effectiveness by naltrexone. - **Option B**: Continuing naltrexone with small doses of morphine. Morphine is also an opioid and would be reduced in effectiveness by naltrexone. - **Option C**: Continuing naltrexone with small doses of nalbuphine. Nalbuphine is less likely to be fully blocked by naltrexone due to its mixed mechanism of action. - **Option D**: Discontinuing naltrexone and treating pain with morphine. Discontinuing naltrexone might not be ideal as it is part of the patient's treatment for alcohol abuse.

Considering the above points, the best option would be to use an analgesic that is less likely to be fully blocked by naltrexone.

**Final Answer: C. Continue naltrexone with small doses of nalbuphine every 4 hours as needed**

**Case 2: Cardiogenic Shock**

**Question:** A 70-year-old man presents to the hospital with chest pain and dyspnea. ECG shows 2-mm ST elevations in V2-V4. His vital signs are as follows: HR 85, BP 100/55, SpO2 95% on room air. Laboratory test results are remarkable for troponin 3.5 ng/mL and lactate 6 mmol/L. Emergent cardiac catheterization is planned. What other treatment should be initiated at this time?
**A.** Dopamine **B.** 1 L crystalloid bolus **C.** Dobutamine **D.** Norepinephrine **E.** No other intervention necessary besides immediate cardiac catheterization
**Correct Answer:** C

**Gemini-2.5-Pro (Incorrect):** This is a step-by-step analysis of the question and options.
1. **Analyze the Patient's Clinical Presentation:** * **Diagnosis:** The patient presents with classic symptoms of an acute myocardial infarction (chest pain, dyspnea). The ECG confirms an ST-segment elevation myocardial infarction (STEMI) in the anterior leads (V2-V4), which typically involves the left anterior descending (LAD) artery. The elevated troponin level further confirms myocardial necrosis. * **Hemodynamics:** The patient is hypotensive with a blood pressure of 100/55 mmHg. The mean arterial pressure (MAP) is approximately 70 mmHg [(2*55 + 100)/3]. While a MAP of 70 might seem borderline acceptable, the systolic pressure is low, and more importantly, there is evidence of end-organ hypoperfusion. * **Evidence of Shock:** The elevated lactate level of 6 mmol/L is a critical finding. It indicates significant tissue hypoperfusion and anaerobic metabolism, which is the hallmark of shock. * **Synthesis:** The combination of a large anterior STEMI (a common cause of pump failure), hypotension, and elevated lactate establishes the diagnosis of cardiogenic shock. The primary problem is the heart's inability to pump effectively, leading to inadequate cardiac output and systemic hypoperfusion.
2. **Evaluate the Treatment Options:** * **A. Dopamine:** Dopamine is a vasopressor and inotrope. However, clinical trials (e.g., the SOAP II trial) have shown that in patients with cardiogenic shock, dopamine is associated with a higher incidence of arrhythmias and potentially higher mortality compared to norepinephrine. Therefore, it is no longer considered the first-line agent for cardiogenic shock. * **B. 1 L crystalloid bolus:** In cardiogenic shock, the primary problem is pump failure, not volume depletion. The left ventricle is already failing and likely has a high filling pressure (left ventricular end-diastolic pressure). Administering a large fluid bolus would further increase this pressure, potentially worsening pulmonary edema (as suggested by the patient's dyspnea) without improving cardiac output, according to the Frank-Starling mechanism in a failing heart. Fluid administration should be done with extreme caution, if at all, in cardiogenic shock. * **C. Dobutamine:** Dobutamine is a primary inotrope (increases contractility) with some vasodilatory effects (due to beta-2 agonism). While it can improve cardiac output, its vasodilatory properties can worsen hypotension, which is dangerous in a patient who is already hypotensive. It is often used after blood pressure has been stabilized with a vasopressor or in patients with cardiogenic shock who are not significantly hypotensive. Starting dobutamine as the initial agent in this patient would be inappropriate. * **D. Norepinephrine:** Norepinephrine is the first-line vasopressor recommended by current guidelines (e.g., European Society of Cardiology, American Heart Association) for the management of cardiogenic shock. It is a potent alpha-1 agonist, which causes peripheral vasoconstriction and increases systemic vascular resistance (SVR), thereby raising blood pressure. It also has modest beta-1 agonist effects, which can provide some inotropic support without causing the significant tachycardia or arrhythmogenicity associated with dopamine. By increasing diastolic blood pressure, it also improves coronary perfusion pressure, which is crucial in the setting of an MI. Initiating a norepinephrine infusion is the most appropriate step to stabilize the patient's blood pressure and improve organ perfusion while awaiting definitive revascularization. * **E. No other intervention necessary besides immediate cardiac catheterization:** While emergent cardiac catheterization is the definitive treatment to address the underlying cause of the shock (the occluded coronary artery), the patient is hemodynamically unstable. Delaying hemodynamic support until arrival in the catheterization lab is unsafe and increases the risk of cardiac arrest and further end-organ damage. The patient must be stabilized pharmacologically as a bridge to revascularization.
**Conclusion:** The patient is in cardiogenic shock secondary to an anterior STEMI. The immediate priority, alongside preparing for emergent revascularization, is to correct the hypotension and improve tissue perfusion. Norepinephrine is the recommended first-line vasopressor for this purpose.
**Final Answer:** The final answer is **D**

**Morpheus-14B (Correct):** First, the patient is a 70-year-old man with chest pain and dyspnea. The ECG shows ST elevations in V2-V4, which suggests an anterior STEMI (ST-elevation myocardial infarction). His vital signs are: HR 85 (which is normal), BP 100/55 (hypotensive, as diastolic is low), SpO2 95% on room air (good oxygenation). Lab results: troponin 3.5 ng/mL (elevated, indicating myocardial injury), lactate 6 mmol/L (high, indicating tissue hypoperfusion or shock).
He's scheduled for emergent cardiac catheterization, which is appropriate for STEMI.
Now, the question is: what other treatment should be initiated at this time? The options are:

A. Dopamine

B. 1 L crystalloid bolus

C. Dobutamine

D. Norepinephrine

E. No other intervention necessary besides immediate cardiac catheterization

I need to think about the patient's condition. He has hypotension (BP 100/55), which is concerning. Hypotension in a STEMI patient can indicate cardiogenic shock or low perfusion.

Lactate is elevated at 6 mmol/L, which is high. Normal lactate is less than 2 mmol/L, so this suggests significant tissue hypoperfusion, likely due to the heart attack.

In STEMI, especially with hypotension and elevated lactate, we need to address the shock state. The goal is to improve perfusion and support the heart.

Let's evaluate each option:

A. Dopamine: Dopamine is a vasopressor that can increase heart rate and contractility at higher doses. It's used in shock states, but it's not first-line for cardiogenic shock from STEMI. Dopamine can have arrhythmogenic effects and isn't preferred in acute coronary syndromes.

B. 1 L crystalloid bolus: This is fluid resuscitation. In hypotension, fluids can be given to increase preload and cardiac output. However, in cardiogenic shock from STEMI, fluid boluses are generally avoided because they can worsen pulmonary edema or increase left ventricular end-diastolic pressure. The patient has dyspnea, which might indicate heart failure or pulmonary issues, so fluids could be harmful.

C. Dobutamine: Dobutamine is a beta-agonist inotropic agent that increases cardiac contractility and heart rate. It's often used in cardiogenic shock to support cardiac output. In STEMI with hypotension, dobutamine can be beneficial to improve perfusion while waiting for reperfusion.

D. Norepinephrine: Norepinephrine is a potent vasopressor that increases systemic vascular resistance and blood pressure. It's commonly used in septic shock or other distributive shocks, but in cardiogenic shock, it can be used if other agents fail. However, it might not be the first choice because it can cause vasoconstriction, which might increase afterload and reduce coronary perfusion.

E. No other intervention necessary: But the patient is hypotensive and has elevated lactate, indicating shock. Immediate catheterization is planned, but we need to stabilize him before or during transport.

The patient is going for emergent catheterization, so interventions should support him until reperfusion. In STEMI management guidelines, for patients with cardiogenic shock, inotropes like dobutamine are recommended to support hemodynamics while PCI is performed.

Also, lactate level of 6 mmol/L is high, suggesting severe shock. We need to address this urgently.

Option B, crystalloid bolus, is not ideal because it could worsen the situation.

Option A, dopamine, is less preferred due to potential side effects.

Option D, norepinephrine, might be used if dobutamine isn't sufficient, but it's not first-line for cardiogenic shock from STEMI.

The most appropriate choice is C, dobutamine, because it directly supports cardiac contractility without increasing afterload excessively.

In some protocols, norepinephrine is used, but dobutamine is more specific for inotropy in heart failure.

The patient has dyspnea, which could be due to pulmonary edema, so inotropes might help by increasing cardiac output and reducing congestion.

Also, hypotension needs correction.

So, I think C. Dobutamine is the best choice.

But let's confirm the BP: 100/55. Systolic is 100, which is low for a 70-year-old, and diastolic is 55, which is very low. This indicates poor perfusion.

In STEMI with hypotension, dobutamine is often initiated to support the heart.

Moreover, lactate is high, so we need to improve perfusion.

Therefore, the answer should be C. Dobutamine.

The patient presents with ST-elevation myocardial infarction (STEMI) confirmed by ECG, hypotension (BP 100/55), elevated troponin (3.5 ng/mL), and significantly elevated lactate (6 mmol/L), indicating cardiogenic shock. The emergent cardiac catheterization is appropriate, but additional interventions are needed to support hemodynamics and improve tissue perfusion before reperfusion.

- **Option A (Dopamine)**: Dopamine is a vasopressor that can increase heart rate and contractility at higher doses but is not first-line for cardiogenic shock in STEMI due to potential arrhythmogenic effects and lack of strong evidence for benefit in this setting. - **Option B (1 L crystalloid bolus)**: Fluid resuscitation is generally avoided in cardiogenic shock from STEMI because it can worsen pulmonary edema or increase left ventricular end-diastolic pressure, exacerbating dyspnea and heart failure. - **Option C (Dobutamine)**: Dobutamine is a beta-agonist inotropic agent that increases cardiac contractility and heart rate, improving cardiac output and perfusion without significantly increasing afterload. It is recommended in guidelines for cardiogenic shock in STEMI to support hemodynamics while awaiting reperfusion. - **Option D (Norepinephrine)**: Norepinephrine is a potent vasopressor that increases systemic vascular resistance and blood pressure but can reduce coronary perfusion by

increasing afterload. It is typically reserved for refractory shock or when inotropes like dobutamine are insufficient. - **Option E (No other intervention necessary)**: The patient's hypotension and elevated lactate indicate shock, requiring immediate intervention to prevent organ damage. Delaying treatment could worsen outcomes.
The most suitable choice is **C. Dobutamine**, as it directly addresses the cardiogenic shock by enhancing cardiac contractility, supporting perfusion, and reducing the risk of complications during transport to catheterization.

**Qwen-2.5-14B-Instruct (Incorrect):** Let's analyze the patient's condition step by step:
1. **Clinical Presentation**: The patient is a 70-year-old male with chest pain and dyspnea, which are classic symptoms of a possible acute coronary syndrome (ACS), specifically a STEMI (ST-elevation myocardial infarction) given the ECG findings of 2-mm ST elevations in V2-V4.
2. **Vital Signs**: HR 85, BP 100/55, SpO2 95% on room air. The blood pressure is notably low (hypotension), which is concerning and suggests possible cardiogenic shock.
3. **Laboratory Results**: Troponin 3.5 ng/mL (elevated, indicating myocardial damage) and lactate 6 mmol/L (elevated, indicating tissue hypoperfusion and possible shock).
4. **Management**: Given the patient's presentation, emergent cardiac catheterization is appropriate. However, the patient's hypotension and elevated lactate suggest the need for immediate hemodynamic support.
Now, let's evaluate each option:
A. **Dopamine**: Dopamine can be used in shock states, but it has a dose-dependent effect. At low doses, it can increase renal and splanchnic blood flow, but at higher doses, it can increase myocardial oxygen demand and potentially worsen ischemia. It is not the first-line agent for hypotension in this setting.
B. **1 L Crystalloid Bolus**: This is a reasonable initial step in hypotensive patients to assess for volume responsiveness. However, given the patient's low blood pressure and elevated lactate, it is likely that the patient is in cardiogenic shock rather than hypovolemic shock, making this less likely to be the primary intervention.
C. **Dobutamine**: This is a beta-1 selective inotrope that increases cardiac output by increasing myocardial contractility. It is often used in cardiogenic shock to improve cardiac output and perfusion. However, it does not address the hypotension directly.
D. **Norepinephrine**: This is a potent alpha-1 agonist that increases systemic vascular resistance and can be used to treat hypotension in cardiogenic shock. It is the most appropriate choice for immediate hemodynamic support in this patient.
E. **No other intervention necessary besides immediate cardiac catheterization**: Given the patient's hypotension and elevated lactate, immediate hemodynamic support is necessary to stabilize the patient before cardiac catheterization.
Considering the patient's hemodynamic instability and the need for immediate support, the most appropriate initial treatment is:
**D. Norepinephrine**
This choice addresses the patient's hypotension and provides the necessary hemodynamic support while preparing for emergent cardiac catheterization.

## F  SIGNIFICANCE TEST

To rigorously validate the claims presented in Sec. 5.1, we conducted a series of statistical tests to confirm the validity and significance of our insights. This section presents the results of these tests and discusses additional findings uncovered during the analysis.

**Methodology**  By default, all comparisons were evaluated using bootstrap resampling. We report the difference in means ($\Delta$), the 90% confidence interval (CI, corresponding to $\alpha = 0.1$), and the one- or two-sided p-value.

### F.1  SIGNIFICANCE TEST ON MODEL CHARACTERISTIC

**Performance Across Model Scale**  Our analysis confirmed the relationship between model scale and performance as described in Sec. 5.1. The claim of a "strong positive correlation between model performance and scale" was strongly supported by Spearman correlation tests across all systems. As detailed in Tab. 13, the Spearman's rank correlation coefficient ($\rho$) exceeds 0.8 for all three systems,

while the corresponding p-values approach zero. These results indicate a strong and statistically significant positive association.

Table 13: Statistical Test for Model Scale

| System | $\rho_s$ | p-value |
|--------|----------|---------|
| System1 | 0.859 | $1.49e^{-15}$ |
| System1.x | 0.861 | $1.07e^{-15}$ |
| System2 | 0.818 | $3.31e^{-16}$ |

Table 14: Statistical Test for Marginal Return

| System | $\beta_2$ | p-value |
|--------|-----------|---------|
| System1 | $-2.79$ | 0.006 |
| System1.x | $-2.607$ | 0.007 |
| System2 | $-1.920$ | 0.008 |

The hypothesis of diminishing marginal returns was also supported. We employed a quadratic ordinary least squares (OLS) model to examine the relationship between model size and performance score. For this analysis, the alternative hypothesis posits that the coefficient of the quadratic term, $\beta_2$, is less than zero. The results of one-sided tests confirmed a statistically significant negative coefficient for this quadratic term across all evaluated systems (all $p < 0.025$), indicating the presence of diminishing returns.

To evaluate the claim that "the slope for System2 is significantly lower than System1.x, indicating that performance gains from increasing model size are markedly smaller for System2," we employed a linear model. The model is defined as:

$$y = \beta_0 + \beta_1 x + \beta_2 \cdot \text{group} + \beta_3(x \times \text{group}) + \varepsilon. \quad (2)$$

In this model, the **group** variable is a binary indicator, specifying whether the data point originated from System1.x or System2. We conducted a one-sided bootstrap hypothesis test with the alternative hypothesis that $\beta_3$ is greater than zero.

Our tests revealed this holds true when comparing System 2 to System 1.x (p=0.0009). In contrast, the difference between System 2 and System 1 (factual recall) was not statistically significant (p=0.1538), suggesting that the scaling advantage diminishes most sharply when moving from hybrid reasoning to complex decision-making.

**Performance Across Language**   Statistical analysis corroborates the cross-lingual observations detailed in Section 5.1. To test the hypothesis that English accuracy exceeds Chinese accuracy ($H_1 : Acc_{\text{EN}} > Acc_{\text{CH}}$), we performed bootstrap tests on the AnesBench Chinese-English mutual translation benchmark. The results strongly indicate that models based on Llama-3.1-8B perform notably worse in Chinese. For all evaluated models in this series, this performance disparity was statistically significant ($p < 0.05$), highlighting that language transferability remains a key challenge.

Table 15: Statistical Test for Performance Across Language

| Model | p-value |
|-------|---------|
| Llama-3.1-8B-Instruct | $9e^{-5}$ |
| Llama-3.1-8B-UltraMedical | $9e^{-5}$ |
| FineMedLM-o1 | $9e^{-5}$ |

**Performance Across Output Length**   Regarding the impact of reasoning length, we tested the claim that "Models with longer CoT reasoning processes tend to exhibit superior response performance." in Section 5.1. A Spearman correlation test provided weak evidence for this relationship in the context of System 2 tasks ($\rho$=0.4917, p=0.0741). No significant correlation was found for System 1 or System 1.x tasks, reinforcing the idea that extended reasoning is primarily beneficial for problems that demand complex, multi-step cognitive processes.

### F.2 SIGNIFICANCE TEST ON TRAINING METHODOLOGY

**Impact of Continuous Pre-Training**   We commenced our investigation by examining the claim from Sec. 5.2 concerning the dichotomous effect of CPT. This involved a comparative anal-

ysis of the top-performing models trained with and without the CPT methodology. Specifically, we formulated distinct alternative hypotheses ($H_1$) for the English and Chinese benchmarks. For AnesBench-English, the hypothesis was $H_1$ : AnesQA + Medical-o1 (with CPT) > AnesQA + Medical-o1 (without CPT), whereas for the AnesBench-Chinese, it was $H_1$ : AnesQA + Medical-o1 (without CPT) > AnesQA + Medical-o1 (with CPT).

To test these hypotheses, we employed a one-sided bootstrap test. The results, as presented in Table 16, indicate that the p-values for both tests are below the 0.05 significance level. Consequently, both alternative hypotheses are accepted. This finding substantiates the initial claim: CPT improves the upper bound of the model's performance on AnesBench-English, while it degrades performance on the AnesBench-Chinese.

Table 16: Statistical Test for the Impact of Continuous Pre-Training

| Subset | $\Delta$ | p-value | 90% CI |
|---|---|---|---|
| AnesBench-English | 0.0149 | 0.0413 | (0.0009, 0.0287) |
| AnesBench-Chinese | 0.0592 | 0.000 09 | (0.0438, 0.0746) |

**Complementarity of AnesQA and Medical-o1** To assess the complementarity between AnesQA and Medical-o1, we conducted four sets of tests under both CPT-trained and non-CPT-trained conditions. These experiments evaluated whether combining AnesQA and Medical-o1 could yield better performance than using either individually. As shown in Tab. 17, a statistically significant improvement was observed only when both models were combined under the CPT-trained setting and evaluated on AnesBench-English. Accordingly, we used the phrase "leads to varying degrees of performance improvement" in the main text to reflect this outcome. Although the performance gains from the combination are not universally significant, consistent improvements are observed regardless of CPT training and across both AnesBench-English and AnesBench-Chinese benchmarks. This consistency suggests that combining the two models generally yields incremental benefits.

Table 17: Overall Complementarity of Supervised Fine-Tuning Datasets

| Comparison | $\Delta$ | p-value | 90% CI |
|---|---|---|---|
| *Without Continuous Pre-Training* | | | |
| AnesQA + Medical-o1 > AnesQA | 0.0041 | 0.2794 | (-0.0079, 0.0149) |
| AnesQA + Medical-o1 > Medical-o1 | 0.0059 | 0.2216 | (-0.0070, 0.0185) |
| *With Continuous Pre-Training* | | | |
| AnesQA + Medical-o1 > AnesQA | 0.0154 | 0.0237 | (0.0029, 0.0278) |
| AnesQA + Medical-o1 > Medical-o1 | 0.0056 | 0.2284 | (-0.0068, 0.0181) |

## G   SUPPLEMENTARY EVALUATION

### G.1   EVALUATION ON OUT OF DISTRIBUTION BENCHMARKS

To further assess the generalization capability of the model trained on anesthesiology reasoning data, we evaluated Morpheus across three general medical benchmarks (MedQA, MedMCQA, and MedExpQA; Jin et al., 2021; Pal et al., 2022; Alonso et al., 2024) and three general domain benchmarks (MMLU, MMLU-Pro, and AGIeval; Wang et al., 2023a; 2024b; Zhong et al., 2024). Consistent with our previous experiments, we employed the same framework used in AnesBench to extract answers for multiple-choice questions. Notably, for AGIeval, we restricted the evaluation to its 18 MCQ tasks, whereas comprehensive evaluations were conducted for all other benchmarks. The medical subset of MMLU includes the tasks anatomy, clinical_knowledge, college_biology, college_medicine, high_school_biology, medical_genetics, nutrition, professional_medicine, and virology.

The Morpheus-7B/14B/32B models, fine-tuned via SFT and GRPO on anesthesia reasoning data, consistently outperform the Qwen2.5-Instruct baselines across all three out-of-distribution general

| Model | AGIEval | MMLU | MMLU-Pro | MedExpQA | MedMCQA | MedQA |
|---|---|---|---|---|---|---|
| Qwen2.5-7B-Instruct | 0.649 | 0.709 | 0.455 | 0.526 | 0.532 | 0.654 |
| Morpheus-7B | 0.663 | 0.786 | 0.580 | 0.582 | 0.593 | 0.720 |
| Qwen2.5-14B-Instruct | 0.701 | 0.801 | 0.627 | 0.642 | 0.593 | 0.735 |
| Morpheus-14B | 0.747 | 0.843 | 0.667 | 0.752 | 0.661 | 0.798 |
| Qwen2.5-32B-Instruct | 0.739 | 0.828 | 0.672 | 0.746 | 0.628 | 0.780 |
| Morpheus-32B | 0.766 | 0.873 | 0.717 | 0.798 | 0.685 | 0.814 |
| Qwen2.5-72B-Instruct | 0.742 | 0.852 | 0.703 | 0.790 | 0.674 | 0.812 |

Table 18: Evaluation Result on OOD Benchmarks

| Model | English | French | Italian | Spanish |
|---|---|---|---|---|
| Qwen2.5-7B-Instruct | 0.600 | 0.512 | 0.512 | 0.480 |
| Morpheus-7B | 0.632 | 0.608 | 0.464 | 0.624 |
| Qwen2.5-14B-Instruct | 0.712 | 0.576 | 0.632 | 0.648 |
| Morpheus-14B | 0.736 | 0.728 | 0.776 | 0.768 |
| Qwen2.5-32B-Instruct | 0.784 | 0.768 | 0.712 | 0.720 |
| Morpheus-32B | 0.784 | 0.808 | 0.808 | 0.792 |
| Qwen2.5-72B-Instruct | 0.800 | 0.776 | 0.776 | 0.808 |

Table 19: Evaluation Result on MedExpQA

medical benchmarks. In certain settings, they even surpass the performance of 72B-scale models, indicating that our training strategy primarily enhances general medical reasoning capabilities rather than merely causing overfitting to the source task. Regarding language transferability, Morpheus generally outperforms the corresponding Qwen2.5-Instruct models across four languages on the MedExpQA benchmark. The improvements are particularly significant in Italian and Spanish, demonstrating strong cross-lingual reasoning transfer. A similar trend is observed in MedQA, where performance gains are notably substantial in Traditional Chinese and English contexts.

In terms of general-domain generalization, Morpheus demonstrates strong performance on benchmarks such as AGIEval, MMLU, and MMLU-Pro. The performance improvements observed across the majority of tasks indicate that the reasoning gains derived from our training data extend beyond the medical domain to general-purpose tasks.

Since we performed no training or fine-tuning on domains other than anesthesiology, no new domain-specific knowledge was introduced. Consequently, these performance gains can be attributed to enhanced reasoning capabilities, allowing the model to more effectively utilize its internal knowledge base. While prior works (Cheng et al., 2025) have focused on multi-domain reasoning generalization, our findings offer new insights into generalization derived from a highly specific, smaller source domain.

## G.2 SELF-CONSISTENCY IN TEST TIME SCALING

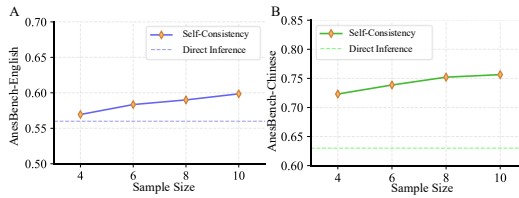

Figure 12: **Test time scaling of Self-Consistency.**

We utilized AnesBench to evaluate the efficacy of test-time scaling techniques, such as Self-Consistency (Wang et al., 2023b), for enhancing reasoning in anesthesiology. Specifically, we employed the Morpheus-7B as the policy model for sampling. As illustrated in Fig. 12, the application of Self-Consistency markedly surpasses the performance of direct inference. Furthermore, the model's performance consistently improves with an increased number of samples. These findings underscore that test-time scaling is an effective method for unlocking the full potential of LLMs in complex anesthesiology reasoning tasks.

| Model | Mainland | Taiwan | US |
|---|---|---|---|
| Qwen2.5-7B-Instruct | 0.797 | 0.660 | 0.505 |
| Morpheus-7B | 0.795 | 0.709 | 0.656 |
| Qwen2.5-14B-Instruct | 0.816 | 0.745 | 0.643 |
| Morpheus-14B | 0.835 | 0.815 | 0.746 |
| Qwen2.5-32B-Instruct | 0.853 | 0.804 | 0.682 |
| Morpheus-32B | 0.831 | 0.817 | 0.794 |
| Qwen2.5-72B-Instruct | 0.861 | 0.833 | 0.742 |

Table 20: Evaluation Result on MedQA

| Model | AQuA-RAT | Gaokao | JEC | LSAT | LogiQA | SAT |
|---|---|---|---|---|---|---|
| Qwen2.5-7B-Instruct | 0.622 | 0.740 | 0.563 | 0.476 | 0.528 | 0.722 |
| Morpheus-7B | 0.721 | 0.737 | 0.575 | 0.489 | 0.530 | 0.766 |
| Qwen2.5-14B-Instruct | 0.752 | 0.767 | 0.601 | 0.533 | 0.621 | 0.798 |
| Morpheus-14B | 0.772 | 0.817 | 0.670 | 0.585 | 0.648 | 0.830 |
| Qwen2.5-32B-Instruct | 0.744 | 0.817 | 0.663 | 0.576 | 0.652 | 0.804 |
| Morpheus-32B | 0.787 | 0.820 | 0.696 | 0.628 | 0.705 | 0.843 |
| Qwen2.5-72B-Instruct | 0.752 | 0.812 | 0.676 | 0.577 | 0.663 | 0.815 |

Table 21: Evaluation Result on AGIeval

| Model | medical | College | High_School | Other | Professional |
|---|---|---|---|---|---|
| Qwen2.5-7B-Instruct | 0.728 | 0.595 | 0.773 | 0.703 | 0.581 |
| Morpheus-7B | 0.782 | 0.734 | 0.859 | 0.773 | 0.673 |
| Qwen2.5-14B-Instruct | 0.797 | 0.724 | 0.864 | 0.796 | 0.693 |
| Morpheus-14B | 0.842 | 0.791 | 0.905 | 0.834 | 0.741 |
| Qwen2.5-32B-Instruct | 0.827 | 0.801 | 0.896 | 0.812 | 0.732 |
| Morpheus-32B | 0.862 | 0.875 | 0.922 | 0.862 | 0.789 |
| Qwen2.5-72B-Instruct | 0.857 | 0.810 | 0.912 | 0.837 | 0.771 |

Table 22: Evaluation Result on MMLU

| Task | Q-7B-It | M-7B | Q-14B-It | M-14B | Q-32B-It | M-32B | Q-72B-It |
|---|---|---|---|---|---|---|---|
| Health | 0.502 | 0.630 | 0.643 | 0.703 | 0.686 | 0.737 | 0.698 |
| Biology | 0.695 | 0.738 | 0.787 | 0.827 | 0.829 | 0.861 | 0.849 |
| Business | 0.459 | 0.649 | 0.710 | 0.693 | 0.734 | 0.771 | 0.769 |
| Chemistry | 0.413 | 0.572 | 0.657 | 0.627 | 0.691 | 0.730 | 0.736 |
| Computer Sci | 0.400 | 0.637 | 0.649 | 0.749 | 0.698 | 0.773 | 0.756 |
| Economics | 0.556 | 0.679 | 0.719 | 0.778 | 0.751 | 0.820 | 0.780 |
| Engineering | 0.366 | 0.339 | 0.490 | 0.396 | 0.550 | 0.474 | 0.542 |
| History | 0.417 | 0.533 | 0.541 | 0.625 | 0.583 | 0.640 | 0.619 |
| Law | 0.299 | 0.326 | 0.341 | 0.445 | 0.406 | 0.501 | 0.465 |
| Math | 0.486 | 0.759 | 0.778 | 0.810 | 0.799 | 0.869 | 0.826 |
| Other | 0.432 | 0.556 | 0.590 | 0.672 | 0.648 | 0.703 | 0.695 |
| Philosophy | 0.361 | 0.453 | 0.529 | 0.629 | 0.581 | 0.657 | 0.611 |
| Physics | 0.400 | 0.590 | 0.657 | 0.642 | 0.719 | 0.728 | 0.739 |
| Psychology | 0.589 | 0.654 | 0.691 | 0.734 | 0.727 | 0.773 | 0.756 |

Table 23: Evaluation Result on MMLU−Pro

Table 24: Analysis results of the model's reasoning hallucinations on the AnesBench-English system2 subset. The numbers in the cells respectively represent the quantity of various types of reasoning sentences in the sampled subset and their proportion of the total reasoning sentences.

| Model | Valid | NS | CR | OE | MC |
|---|---|---|---|---|---|
| **AnesBench-English System2** | | | | | |
| Morpheus-7B | 953 (76.2%) | 206 (16.5%) | 13 (1.0%) | 66 (5.3%) | 12 (1.0%) |
| Qwen2.5-7B-Instruct | 181 (68.8%) | 55 (20.9%) | 1 (0.4%) | 18 (6.8%) | 8 (3.0%) |
| Morpheus-14B | 1119 (81.4%) | 191 (13.9%) | 5 (0.4%) | 51 (3.7%) | 8 (0.6%) |
| Qwen2.5-14B-Instruct | 254 (77.0%) | 51 (15.5%) | 4 (1.2%) | 18 (5.5%) | 3 (0.9%) |
| Morpheus-32B | 1179 (83.6%) | 151 (10.7%) | 4 (0.3%) | 62 (4.4%) | 14 (1.0%) |
| Qwen2.5-32B-Instruct | 233 (78.2%) | 43 (14.4%) | 1 (0.3%) | 15 (5.0%) | 6 (2.0%) |
| Qwen2.5-72B-Instruct | 230 (74.9%) | 52 (16.9%) | 5 (1.6%) | 13 (4.2%) | 7 (2.3%) |

*Notes: **NS**: Non-Sequitur, **CR**: Causal Reversal, **OE**: Over-extrapolation, **MC**: Medical Contraindication.*

### G.3 HALLUCINATION EVALUATION

Beyond the accuracy on MCQs, ensuring the accuracy of each reasoning step generated by the model is of greater importance when translating research findings into practical clinical applications. Therefore, we performed a detailed hallucination evaluation on the outputs of the several models presented in Tab. 4, utilizing an LLM-as-Judge paradigm.

Specifically, our analytical pipeline comprises several steps. First, we sampled 50 questions from System 2 of the English subsets of AnesBench. For these questions, we employed spaCy [4] to segment each model's response into independent sentences. Subsequently, the context and each individual sentence were fed into GPT-OSS-120B (Agarwal et al., 2025) for coreference resolution, while simultaneously filtering for complete sentences that contained explicit reasoning behaviors. Finally, GPT-OSS-120B, together with an evaluation prompt, was used to assess each sentence for the presence of four common reasoning-hallucination patterns during inference: Non-Sequitur, Causal Reversal, Medical Contraindication, and Over-extrapolation.

As shown in Tab. 24, the results indicate that among these hallucination types, Non-Sequitur and Over-extrapolation are the most prevalent. This suggests that models are highly susceptible to broken logical chains or drawing conclusions based on insufficient evidence, highlighting the need for future research to employ methods such as process reward model (Lightman et al., 2023) and retrieval-augmented generation (Lewis et al., 2020; Jiang et al., 2025) to enhance reasoning coherence and mitigate issues of insufficient evidence. Moreover, scaling these retrieval systems to accommodate large and multimodal clinical data will be increasingly critical (Müller & Unay, 2017; Shen et al., 2022; 2024b;a).

Comparing the models, a larger parameter scale generally correlates with stronger reasoning validity in most cases. A direct comparison between the Morpheus and Qwen2.5 models reveals that Morpheus tends to generate more extensive reasoning steps. Notably, Morpheus demonstrated a consistent improvement in reasoning validity over the baseline Qwen2.5, thereby substantiating the effectiveness of our training data and methodology. Furthermore, across various parameter scales, the Morpheus models consistently maintain the MC (high-risk hallucination) error rate at or below 1.0%, which likewise demonstrates the enhanced safety profile of the models trained with our resources.

> **Prompt used for Hallucination Evaluation**
>
> **SYSTEM PROMPT**

---

[4] spaCy: https://spacy.io/

```
You are an expert Clinical Logic Judge. Your task is to evaluate
    ↪ the logical soundness of a single reasoning step generated by
    ↪  a medical AI.
You are NOT verifying isolated facts; you are judging the causal
    ↪ link and logical deduction within the sentence. Does the
    ↪ stated premise or fact logically justify the clinical
    ↪ conclusion or recommendation?
# EVALUATION CRITERIA:
Evaluate the internal logic of the Target_Sentence for the
    ↪ following fatal flaws:
- Non-Sequitur: The conclusion does not logically follow from the
    ↪ premise (e.g., "Patient has a headache, therefore amputate
    ↪ the leg").
- Causal Reversal: Mixing up cause and effect.
- Medical Contraindication: Recommending a treatment that is
    ↪ logically contradicted by the premise stated in the very same
    ↪  sentence (e.g., "Given the severe bleeding, administer
    ↪ heparin").
- Over-extrapolation: Jumping to a definitive, extreme conclusion
    ↪ without sufficient logical bridging.
# EVALUATION CATEGORIES:
1. "Valid": The reasoning is medically sound, coherent, and the
    ↪ conclusion logically follows the premises provided in the
    ↪ sentence.
2. "Flawed": The reasoning contains a logical fallacy,
    ↪ contradiction, or medically dangerous leap in logic.
# INSTRUCTIONS:
Analyze the "Target_Sentence". Output ONLY a valid JSON object with
    ↪  the following structure:
{
  "logic_judgment": "<Must be exactly one of: Valid, Flawed>",
  "flaw_type": "<If Flawed, briefly name the issue (e.g., Non-
      ↪ Sequitur, Contradiction). If Valid, output null>",
  "rationale": "Briefly explain why the logical bridge holds up or
      ↪ breaks down."
}
# EXAMPLES:
Target_Sentence: "Because the patient is allergic to penicillin, we
    ↪  should prescribe amoxicillin."
Output: {"logic_judgment": "Flawed", "flaw_type": "Contradiction",
    ↪ "rationale": "Amoxicillin is a penicillin-class antibiotic,
    ↪ so the premise of allergy directly contradicts the treatment
    ↪ recommendation."}
Target_Sentence: "Given the elevated troponin levels and chest pain
    ↪ , an ECG should be performed to evaluate for myocardial
    ↪ infarction."
Output: {"logic_judgment": "Valid", "flaw_type": null, "rationale":
    ↪  "Elevated troponin and chest pain are classic signs of
    ↪ cardiac injury, making an ECG a logical next diagnostic step
    ↪ ."}
```

## G.4   EVALUATION ON THE OPEN-ENDED SUBSET

To address the limitations of MCQs and further validate our findings, we augmented our evaluation with open-ended questions. A subset of System 2 benchmark items was converted into an open-ended format.

To construct our experimental dataset, we manually curated a specific subset from System2 questions of AnesBench-English. We identified and selected 64 items that were "option-independent," meaning they could be answered based solely on the information provided in the question stem, without reliance on the accompanying options. Subsequently, we employed the Gemini-2.5-Pro (Comanici et al., 2025) to transform this curated subset of questions into an open-ended format. This conver-

sion was guided by a specific prompt designed to minimize unnecessary alterations to the original questions.

To ensure robust and objective scoring, we use automated metrics such as BLEU and F1 along with LLM-as-judge to evaluate performance on open-ended questions. To implement LLM-as-judge, we engineered a detailed prompt that instructed the evaluator models (GPT-4.1 and Gemini-2.5-Pro) to adopt the persona of an expert medical evaluator. To standardize the assessment, the prompt included a detailed 10-point scoring rubric, mapping score ranges to specific quality levels. We repeated each evaluation three times. The final scores for each response were then averaged across both evaluator models to ensure robust and reliable assessment. The prompts used for LLM-as-Judge and question rewriting are provided as below.

The results presented in Tab. 25 reveal a strong correlation between the LLM-based scores (GPT and Gemini) and the original MCQ accuracy on AnesBench, which validates the use of MCQs as a reliable proxy for complex reasoning. In contrast, automated metrics like BLEU and F1 scores were consistently low (mostly below 0.25), underscoring their inadequacy for assessing the nuanced, multi-step logic required in medical decision-making. Therefore, we prioritize MCQs in our benchmark as a more robust and standardized method for evaluating reasoning performance.

---

**Prompt used for LLM-as-judge**

**SYSTEM PROMPT**

```
You are an expert medical evaluator tasked with scoring responses
    ↪ to medical questions. Your job is to evaluate how well a
    ↪ model's response matches the target answer for medical
    ↪ scenarios.
Evaluation Criteria:
1. Accuracy: Does the response provide the correct medical
    ↪ information that aligns with the target answer?
2. Completeness: Does the response address all key aspects of the
    ↪ question?
3. Clinical Reasoning: Is the medical reasoning sound and
    ↪ appropriate?
4. Clarity: Is the response clear and well-structured?
Scoring Scale: 0-10 points
- 10: Perfect match with target, excellent reasoning, complete and
    ↪ accurate
- 8-9: Very good match with target, minor gaps or slight
    ↪ inaccuracies
- 6-7: Good match with target, some important information missing
    ↪ or unclear
- 4-5: Partial match with target, significant gaps or inaccuracies
- 2-3: Poor match with target, major errors or missing key
    ↪ information
- 0-1: Completely incorrect or irrelevant response
Instructions:
1. First, provide your reasoning and justification for the score
2. Compare the model's response with the target answer
3. Identify strengths and weaknesses in the response
4. Then provide your final score in the exact format: \"Score: X
    ↪ /10\"
Input Format:
- Question: [The medical question/scenario]
- Target Answer: [The correct answer]
- Model Response: [The model's response to evaluate]
Please evaluate the response thoroughly and provide your score.
```

---

**Prompt used for Question Rewriting**

```
You are a converter that rewrites multiple-choice questions (MCQs)
    ↪ into open-ended, directly answerable questions (open QA) with
    ↪  the smallest possible changes to the original stem.
GOAL
- Keep the background/stem text virtually unchanged (only tiny
    ↪ grammar/clarity fixes if absolutely necessary).
- Replace the MCQ-style query ("Which of the following...", options
    ↪  A-E, etc.) with a direct question that can be answered
    ↪ solely from the stem-without referring to any options.
- Do NOT add new facts that aren't already in the stem. Do NOT
    ↪ include answer choices or mention "options."
- Preserve technical terms, numbers, units, time courses, and
    ↪ qualifiers (e.g., "best next step," "initial management," "
    ↪ contraindicated," "EXCEPT/NOT" -> phrase as "Which [thing] is
    ↪  contraindicated/not true in this scenario?" but still as a
    ↪ direct, standalone question).
- Use the same language as the input stem.
- Do NOT introduce or reintroduce "most" qualifiers (e.g., avoid "
    ↪ most likely").
ALLOWED MICRO-EDITS
- Remove phrases like "of the following," "choose," "select,"
    ↪ letters (A/B/C...), and references to options.
- Minimal rewording of the final question clause to be a direct
    ↪ inquiry (e.g., "What medication should be administered next
    ↪ ?", "What is the diagnosis?", "Which intervention should be
    ↪ performed next?").
- If the stem relies on a test or figure, you may add at most one
    ↪ short bridging clause drawn from the stem (e.g., "Based on
    ↪ this finding, ...")-no new information.
OUTPUT FORMAT (STRICT)
- Return EXACTLY one block:
  <refined>YOUR SINGLE, OPEN-ENDED QUESTION HERE</refined>
- No explanations, no option lists, no reasoning, no extra tags or
    ↪ text outside <refined>...</refined>.
One-shot example:
- Input (MCQ with known answer provided for your understanding-do
    ↪ not print it):
    A patient is receiving blood product transfusions after
        ↪ undergoing aortic valve replacement. Despite the
        ↪ transfusions, the surgeons still note that their field is
        ↪  very "oozy." TEG is performed, which reveals a decreased
        ↪  alpha angle. Which blood product should be administered
        ↪ next?
- Correct answer (for reference only): Cryoprecipitate
- Output (note the minimal edits and the required tags):
    <refined>A patient is receiving blood product transfusions
        ↪ after undergoing aortic valve replacement. Despite the
        ↪ transfusions, the surgical field remains very oozy. TEG
        ↪ shows a decreased alpha angle. Based on this finding,
        ↪ what blood product should be administered next?</refined>
VALIDATION CHECKLIST
- [ ] Background preserved with only minimal edits.
- [ ] Final question is direct and answerable from the stem alone,
    ↪ with no reference to options.
- [ ] No new clinical facts added; all original data retained.
- [ ] No "most" qualifiers introduced.
- [ ] Exactly one <refined>...</refined> block returned; nothing
    ↪ else.
THE QUESTION I NEED YOU TO CONVERT IS:
{question}
CORRECT ANSWER (FOR REFERENCE ONLY): {correct_answer}
```

Table 25: Evaluation Results of Open-Ended QA vs. AnesBench (MCQ)

| Model | LLM-based Scores | | Automated Metrics | | AnesBench |
|---|---|---|---|---|---|
| | GPT Score | Gemini Score | BLEU | F1 | Acc (%) |
| DeepSeek-R1 | 8.22 | 8.76 | 0.037 | 0.230 | 82 |
| Gemini-2.5-flash | 7.88 | 8.53 | 0.026 | 0.196 | 81 |
| DeepSeek-V3 | 7.85 | 8.34 | 0.037 | 0.230 | 73 |
| GPT-4o | 7.77 | 7.65 | 0.029 | 0.205 | 77 |
| Gemma-3-27b-it | 6.55 | 6.09 | 0.000 | 0.000 | 58 |
| Qwen3-30B-A3B | 6.44 | 6.02 | 0.000 | 0.000 | 68 |
| DeepSeek-R1-Distill | 6.26 | 5.79 | 0.000 | 0.020 | 63 |
| Qwen2.5-72B-Instruct | 6.17 | 5.51 | 0.000 | 0.000 | 67 |
| Qwen3-14B | 6.10 | 5.45 | 0.000 | 0.068 | 65 |
| Claude-3.7 | 5.11 | 5.35 | 0.020 | 0.165 | 77 |

## G.5 EVALUATION ON CAB

In addition to AnesBench, we evaluated the models on a supplementary resource of over 8,000 MCQs from the CAB (Zhou et al., 2025). The complete evaluation results are presented in Tab. 26. Although this benchmark effectively highlights performance discrepancies among different models, its lack of a structured design precludes a detailed analysis of model capabilities across various systems. Furthermore, the exclusive inclusion of Chinese questions restricts its applicability for multilingual assessment.

Table 26: **Evaluation Results on MCQs from CAB**

| Model | Acc. | Model | Acc. |
|---|---|---|---|
| Baichuan2-7B-Chat | 0.43 | Baichuan2-13B-Chat | 0.48 |
| Bio-Medical-Llama-3-8B | 0.48 | BioMistral-7B | 0.26 |
| chatglm3-6b | 0.36 | Citrus1.0-llama-70B | 0.71 |
| deepseek-r1 | 0.87 | DeepSeek-R1-Distill-Llama-70B | 0.64 |
| DeepSeek-R1-Distill-Qwen-1.5B | 0.26 | DeepSeek-R1-Distill-Qwen-7B | 0.33 |
| DeepSeek-R1-Distill-Qwen-14B | 0.63 | DeepSeek-R1-Distill-Qwen-32B | 0.67 |
| deepseek-v3 | 0.78 | FineMedLM | 0.31 |
| FineMedLM-o1 | 0.35 | gemma-2-9b-it | 0.52 |
| gemma-2-27b-it | 0.56 | gemma-3-1b-it | 0.26 |
| gemma-3-4b-it | 0.43 | gemma-3-12b-it | 0.58 |
| gemma-3-27b-it | 0.65 | glm-4-9b-chat | 0.61 |
| gpt-4o | 0.78 | HuatuoGPT-o1-7B | 0.70 |
| HuatuoGPT-o1-8B | 0.56 | HuatuoGPT-o1-70B | 0.70 |
| HuatuoGPT-o1-72B | 0.81 | internlm3-8b-instruct | 0.84 |
| Llama-3-70B-UltraMedical | 0.71 | Llama-3.1-8B-Instruct | 0.53 |
| Llama-3.1-8B-UltraMedical | 0.54 | Llama-3.3-70B-Instruct | 0.68 |
| Llama-4-Scout-17B-16E-Instruct | 0.78 | Llama3-OpenBioLLM-8B | 0.25 |
| Llama3-OpenBioLLM-70B | 0.64 | Meta-Llama-3-8B-Instruct | 0.49 |
| phi-4 | 0.57 | Qwen2.5-7B-Instruct | 0.67 |
| Qwen2.5-14B-Instruct | 0.73 | Qwen2.5-32B-Instruct | 0.76 |
| Qwen2.5-72B-Instruct | 0.81 | Qwen3-0.6B | 0.37 |
| Qwen3-1.7B | 0.53 | Qwen3-4B | 0.53 |
| Qwen3-8B | 0.65 | Qwen3-14B | 0.76 |
| Qwen3-30B-A3B | 0.73 | Qwen3-32B | 0.80 |
| Qwen3-235B-A22B | 0.75 | QwQ-32B-Preview | 0.73 |
| Yi-1.5-34B-Chat | 0.65 | | |

# H DATASHEET FOR ANESSUITE

MOTIVATION

**1. For what purpose was the dataset created? Was there a specific task in mind? Was there a specific gap that needed to be filled? Please provide a description.**

**A1:** The AnesSuite dataset was created to address a critical gap in artificial intelligence research: the limited exploration of reasoning abilities in LLMs within the highly specialized field of anesthesiology. Its primary goal is to serve as the first comprehensive resource specifically designed to train and evaluate LLMs on tasks relevant to anesthesiology. AnesSuite includes the AnesBench benchmark, which assesses model performance across a wide range of reasoning challenges—from straightforward factual recall to complex clinical decision-making. Additionally, it offers dedicated training datasets that support various strategies for model improvement, including continuous pretraining, supervised fine-tuning, and reinforcement learning.

**2. Who created this dataset (e.g., which team, research group) and on behalf of which entity (e.g., company, institution, organization)?**

**A2:** This dataset is created by the authors of this paper.

**3. Who funded the creation of the dataset? If there is an associated grant, please provide the name of the grantor and the grant name and number.**

**A3:** N/A.

COMPOSITION

**1. What do the instances that comprise the dataset represent (e.g., documents, photos, people, countries)? Are there multiple types of instances(e.g., movies, users, and ratings; people and interactions between them; nodes and edges)? Please provide a description.**

**A1:** The dataset suite's instances include four types: (a) MCQs that assess anesthesiology reasoning at three cognitive levels (System 1, 1.x, 2) in AnesBench; (b) plain-text anesthesiology documents for continued pre-training in AnesCorpus; (c) QA pairs in AnesQA; and (d) verifiable MCQs paired with detailed CoT reasoning in AnesR1.

**2. How many instances are there in total (of each type, if appropriate)?**

**A2:** Instance counts: AnesBench has 4k English MCQs and 3k Chinese MCQs; AnesCorpus contains 1.8 million English and 0.6 million Chinese documents; AnesQA has 20k English QA pairs; AnesR1 totals 10k items (3.2k English + 7k Chinese) of MCQs with CoT.

**3. Does the dataset contain all possible instances or is it a sample (not necessarily random) of instances from a larger set? If the dataset is a sample, then what is the larger set? Is the sample representative of the larger set (e.g., geographic coverage)? If so, please describe how this representativeness was validated/verified. If it is not representative of the larger set, please describe why not (e.g., to cover a more diverse range of instances, because instances were withheld or unavailable).**

**A3:** AnesSuite is a sample drawn from a larger set of instances. AnesBench and AnesR1 are MCQs related to anesthesiology, but they do not imply full representativeness of all anesthesiology questions. AnesCorpus is a keyword-filtered sample from large web corpora; it is therefore a targeted subset, not a census of all anesthesiology texts. AnesQA questions were generated from cleaned PubMed papers via an LLM pipeline with further filtering, which can't imply full representativeness of all anesthesiology QAs.

**4. What data does each instance consist of? "Raw" data (e.g., unprocessed text or images)or features? In either case, please provide a description.**

**A4:** Per-instance data: AnesBench items are MCQs annotated with a cognitive-demand level (System 1/1.x/2). AnesCorpus instances are plain-text documents relevant to anesthesia and pain management. AnesQA instances consist of a question and its answer, with each question typed into one

of five categories. AnesR1 instances pair a verifiable MCQ with a full CoT reasoning trace under the same three-level cognitive framework.

**5. Is there a label or target associated with each instance? If so, please provide a description.**

**A5:** Yes, MCQ datasets (AnesBench, AnesR1) include the correct choice as the target.

**6. Is any information missing from individual instances? If so, please provide a description, explaining why this information is missing (e.g., because it was unavailable). This does not include intentionally removed information, but might include, e.g., redacted text.**

**A6:** No.

**7. Are relationships between individual instances made explicit (e.g., users' movie ratings, social network links)? If so, please describe how these relationships are made explicit.**

**A7:** No.

**8. Are there recommended data splits (e.g., training, development/validation, testing)? If so, please provide a description of these splits, explaining the rationale behind them.**

**A8:** Yes, as mentioned in the main text, we recommend using AnesBench for testing, and AnesCorpus, AnesQA, and AnesR1 for training. There is no explicit validation set, and users can split it from the training set by themselves.

**9. Are there any errors, sources of noise, or redundancies in the dataset? If so, please provide a description.**

**A9:** Yes, since AnesSuite contains datasets from different sources, for example, AnesCorpus is derived from cleaned web data, but it may still contain noise. The construction of AnesQA relies on an LLM pipeline, which may also contain hallucinations and noise. However, each part has corresponding dataset validation methods, as detailed in the relevant sections of the main text and appendices.

**10. Is the dataset self-contained, or does it link to or otherwise rely on external resources (e.g., websites, tweets, other datasets)? If it links to or relies on external resources, a) are there guarantees that they will exist, and remain constant, over time; b) are there official archival versions of the complete dataset (i.e., including the external resources as they existed at the time the dataset was created); c) are there any restrictions (e.g., licenses, fees) associated with any of the external resources that might apply to a future user? Please provide descriptions of all external resources and any restrictions associated with them, as well as links or other access points, as appropriate.**

**A10:** Yes, AnesSuite is self-contained, and users can obtain it directly from the project homepage, HuggingFace, and other channels once the dataset is made public.

**11. Does the dataset contain data that might be considered confidential (e.g., data that is protected by legal privilege or by doctorpatient confidentiality, data that includes the content of individuals non-public communications)? If so, please provide a description.**

**A11:** No, although AnesSuite contains abstract clinical scenarios, they are all virtualized and do not include any personal information.

**12. Does the dataset contain data that, if viewed directly, might be offensive, insulting, threatening, or might otherwise cause anxiety? If so, please describe why.**

**A12:** No.

COLLECTION PROCESS

**1. How was the data associated with each instance acquired? Was the data directly observable (e.g., raw text, movie ratings), reported by subjects (e.g., survey responses), or indirectly inferred/derived from other data (e.g., part-of-speech tags, model-based guesses for age or language)? If data was reported by subjects or indirectly inferred/derived from other data, was the data validated/verified? If so, please describe how.**

**A1:** Yes, in AnesSuite, except for AnesQA, all other data instances can be acquired directly. AnesQA is derived from the LLM pipeline, and its verification process is detailed in the relevant sections of the main text and appendices.

**2. What mechanisms or procedures were used to collect the data (e.g., hardware apparatus or sensor, manual human curation, software program, software API)? How were these mechanisms or procedures validated?**

**A2:** The collection of AnesSuite mainly uses Python scripts. After obtaining the raw data, we adopted both manual and automated verification methods, as detailed in the relevant sections of the main text and appendices.

**3. If the dataset is a sample from a larger set, what was the sampling strategy (e.g., deterministic, probabilistic with specific sampling probabilities)?**

**A3:** The sampling of AnesSuite is a deterministic process, not based on specific sampling probabilities. For example, due to the keyword filtering method, AnesCorpus is only a sample of a deterministic subset related to these keywords among all anesthesia-related texts.

**4. Who was involved in the data collection process (e.g., students, crowdworkers, contractors) and how were they compensated (e.g., how much were crowdworkers paid)?**

**A4:** The data collection was primarily conducted by the lead authors of this paper. Additionally, Tao Huang and Zhiwei Wang from the School of Computer Science, Wuhan University, assisted in the data collection process.

**5. Over what timeframe was the data collected? Does this timeframe match the creation timeframe of the data associated with the instances (e.g., recent crawl of old news articles)? If not, please describe the timeframe in which the data associated with the instances was created.**

**A5**: The collection period of AnesSuite is from 2024 to 2025, which does not exactly match the creation time of the data.

PREPROCESSING/CLEANING/LABELING

**1. Was any preprocessing/cleaning/labeling of the data done (e.g., discretization or bucketing, tokenization, part-of-speech tagging, SIFT feature extraction, removal of instances, processing of missing values)? If so, please provide a description. If not, you may skip the remainder of the questions in this section.**

**A1:** Yes, labeling and cleaning we performed on the data can refer to the relevant sections in the main text and appendices.

**2. Was the "raw" data saved in addition to the preprocessed/cleaned/labeled data (e.g., to support unanticipated future uses)? If so, please provide a link or other access point to the "raw" data.**

**A2:** No.

**3. Is the software used to preprocess/clean/label the instances available? If so, please provide a link or other access point.**

**A3:** All of our preprocessing/cleaning/labeling are done using Python scripts, not a specific software.

USES

**1. Has the dataset been used for any tasks already? If so, please provide a description.**

**A1:** Yes, we have used AnesBench from AnesSuite to evaluate the reasoning abilities of current state-of-the-art LLMs. Additionally, using its training set, we have developed Moupheus, the first collection of reasoning LLMs in anesthesiology.

**2. Is there a repository that links to any or all papers or systems that use the dataset? If so, please provide a link or other access point.**

**A2:** N/A.

**3. What (other) tasks could the dataset be used for?**

**A3:** AnesSuit contains three training sets, among which AnesR1 and AnesQA also include instance labels. This design allows users to freely combine and train models according to their needs. Anes-Bench can also be used to test other LLMs.

**4. Is there anything about the composition of the dataset or the way it was collected and preprocessed/cleaned/labeled that might impact future uses? For example, is there anything that a future user might need to know to avoid uses that could result in unfair treatment of individuals or groups (e.g., stereotyping, quality of service issues) or other undesirable harms (e.g., financial harms, legal risks) If so, please provide a description. Is there anything a future user could do to mitigate these undesirable harms?**

**A4:** No.

**5. Are there tasks for which the dataset should not be used? If so, please provide a description.**

**A5:** N/A

DISTRIBUTION

**1. Will the dataset be distributed to third parties outside of the entity (e.g., company, institution, organization) on behalf of which the dataset was created? If so, please provide a description.**

**A1:** Yes, the dataset will be made available to the public after the paper is published.

**2. How will the dataset will be distributed (e.g., tarball on website, API, GitHub)? Does the dataset have a digital object identifier (DOI)?**

**A2:** AnesSuite will be distributed via GitHub and HuggingFace.

**3. When will the dataset be distributed?**

**A3:** The dataset will be distributed once the paper is accepted after peer review.

**4. Will the dataset be distributed under a copyright or other intellectual property (IP) license, and/or under applicable terms of use (ToU)? If so, please describe this license and/or ToU, and provide a link or other access point to, or otherwise reproduce, any relevant licensing terms or ToU, as well as any fees associated with these restrictions.**

**A4:** For the specific license, please refer to the project homepage and Huggingface after we open-source the AnesSuite.

**5. Have any third parties imposed IP-based or other restrictions on the data associated with the instances? If so, please describe these restrictions, and provide a link or other access point to, or otherwise reproduce, any relevant licensing terms, as well as any fees associated with these restrictions.**

**A5:** No.

**6. Do any export controls or other regulatory restrictions apply to the dataset or to individual instances? If so, please describe these restrictions, and provide a link or other access point to, or otherwise reproduce, any supporting documentation.**

**A6:** No.

MAINTENANCE

**1. Who will be supporting/hosting/maintaining the dataset?**

**A1:** The authors.

**2. How can the owner/curator/manager of the dataset be contacted (e.g., email address)?**

**A2:** Users can contact the authors via the email address on the first page of the main text.

**3. Is there an erratum? If so, please provide a link or other access point.**

**A3:** After the paper is published, we will make public the link pointing to the erratum.

**4. Will the dataset be updated (e.g., to correct labeling errors, add new instances, delete instances)? If so, please describe how often, by whom, and how updates will be communicated to users (e.g., mailing list, GitHub)?**

**A4:** Yes, dataset will be updated. We welcome feedback from AnesSuite users. The authors of this paper will update the dataset on the project homepage and GitHub from time to time based on the feedback received, including removing any errors identified.

**5. Will older versions of the dataset continue to be supported/hosted/maintained? If so, please describe how. If not, please describe how its obsolescence will be communicated to users.**

**A5:** No, each time we update the dataset, we will mark the update time simultaneously, which will help users understand when each version of the dataset becomes obsolete.

**6. If others want to extend/augment/build on/contribute to the dataset, is there a mechanism for them to do so? If so, please provide a description. Will these contributions be validated/verified? If so, please describe how. If not, why not? Is there a process for communicating/distributing these contributions to other users? If so, please provide a description.**

**A6:** N/A.

# I   USAGE OF LLM

**Writing Assistance**   We use LLMs to assist with the proofreading and refinement of the manuscript, primarily for correcting grammar, optimizing sentence structure, and improving the clarity of the language. All content generated with AI assistance was strictly reviewed and revised by the authors to ensure its accuracy and academic integrity. We take full responsibility for the final content of the manuscript.

**Research Applications**   Beyond assisting writing, LLMs are integral to our research. Specifically, we utilized LLMs in the process of dataset construction, evaluation, training and so on. The specific methodologies, implementation details, and the function of these applications in our research are described in detail within the corresponding sections of this paper.

