# OpenReview forum: "AnesSuite: A Comprehensive Benchmark and Dataset Suite for Anesthesiology Reasoning in LLMs"
_ICLR.cc/2026/Conference — ICLR 2026 Poster_

### Official Review · Reviewer_eXNB · 2025-10-30

**Soundness:** 4
**Presentation:** 3
**Contribution:** 2
**Rating:** 6
**Confidence:** 4

**Summary:**

This paper introduces a comprehensive suite for analyzing, training, and evaluating LLMs’ capabilities in the domain of anesthesiology. The benchmark results demonstrate that current models, especially open-weight solutions may not perform perfectly on the benchmark. And both Qwen2.5 models and Morpheus would improve after being trained on the provided datasets.

**Strengths:**

1. The first suite for benchmarking, training, and analyzing failure patterns in the particular domain of anesthesiology
2. Comprehensive analysis and vivid figures demonstrating model performance gaps and some insights on the benchmark

**Weaknesses:**

1. The benchmark is not challenging enough. Overall the best open-weight model could solve 82% of the problems (DeepSeek-R1) and even a smaller Qwen3-14B could solve most of them (70%). These observations indicate that these problems in the benchmark are not challenging enough, especially considering the form of MCQA. One solution may be to refactor the MCQA questions into the free-end generation task that asks the model to answer the question without providing options.
2. Training data effectiveness is not properly demonstrated: table 4 presents models trained on the proposed training set. However, this type of training-testing paradigm falls back to the in-distribution evaluation. A better way to validate the training data would be mixing them with other post-processing data and test the model or test the trained model on other OOD medical tasks.

**Questions:**

How would the model trained on the proposed dedicated data perform on other medical tasks or even general tasks?

Overall, I appreciate the effort of constructing this suite for such a specific domain, it would potentially be a better paper with further clarifications and improvement. I would consider increasing my score once most of my concerns are addressed.

---

> ### Author Response · Authors · 2025-11-21
> **Response to Reviewer eXNB:**
>
> We appreciate your valuable feedback, which has allowed us to strengthen our arguments. We have responded to your specific comments as follows.
>
> ---
>
> ## Response to Weakness 1:  Challenge of Benchmark
>
> We sincerely appreciate your constructive feedback. Regarding your concerns that the benchmark may "lack sufficient challenge" and your suggestion to "introduce open-ended generation tasks," we offer the following response:
>
> 1. Challenge and Discrimination: The higher overall score is primarily driven by system 1 (intuitive, fast thinking) questions. However, this does not mean the benchmark lacks potential challenges.
> 	1. Complex reasoning remains challenging: On the critical System 2 subset, even SOTA open-source models (e.g., DeepSeek-R1) achieve accuracies of only 60%–70%, with the vast majority of models falling below 50%.
> 	2. Broad discriminative power: As illustrated in Table 9, apart from top-tier models, a substantial number of LLMs still struggle with fundamental anesthesiology questions. This stratified design ensures that AnesBench maintains distinct discriminative power across a wide spectrum of model capabilities.
> 	3. The Clinical "Reality Gap": Considering the rigorous standards and near-zero tolerance for error in medical settings, a significant gap remains. From a clinical application perspective, even leading models like Gemini-2.5-Pro and DeepSeek-R1 achieve System 2 scores (approximately 0.6–0.77) that fall far short of the reliability required for real-world deployment. AnesBench delineates the "capability boundaries" of current LLMs in specialized medical reasoning, validating its value as a challenging and rigorous benchmark.
>
> 2.  Validation of Open-Ended Tasks
> We concur with your suggestion and have conducted preliminary experiments detailed in the **Appendix G.2** . We filtered for "option-agnostic" questions within the System 2 subset, restructuring them into open-ended QA tasks. We employed "LLM-as-a-Judge," alongside BLEU and F1 scores, as evaluation metrics. Our findings are as follows:
> 	1. Increased Difficulty: Open-ended generation is significantly more difficult than MCQs, with the majority of models exhibiting weak performance on generation metrics. Notably, for non-top-tier LLMs, automated metrics like BLEU and F1 are extremely low, further highlighting the deficiencies of current LLMs in anesthesiology reasoning tasks.
> 	2. Validity of MCQs: Accuracies of MCQs demonstrate a significant positive correlation with both automated evaluation results and LLM-based scoring. This suggests that, at the current stage, MCQs remain a reliable proxy metric for assessing complex medical reasoning capabilities.
>
> ## Response to Weakness 2 & Question:
>
> We appreciate you and reviewer EdCA for raising this important question regarding the model's generalization capabilities. We systematically evaluated Morpheus on six OOD benchmarks. These covered both general medical contexts (MedQA, MedMCQA, MedExpQA) and general domains (MMLU, MMLU-Pro, AGIEval).
>
> Our experimental results indicate that SFT+GRPO fine-tuning on anesthesia data not only avoided overfitting but significantly enhanced the model's general reasoning capabilities. We will update the manuscript to include the following key findings:
> 1. Out-of-Distribution Medical Tasks: Morpheus-7B/14B outperformed the Qwen2.5-Instruct baseline of the same scale across all three general medical benchmarks. In some cases, they even surpassed the performance of 32B models. Furthermore, the model exhibited significant cross-lingual transfer capabilities, showing consistent improvements across French as well as Spanish and Traditional Chinese.
> 2. Generalization to General Tasks: On general benchmarks such as MMLU and AGIEval, the model's overall performance also surpassed the baselines. This suggests that domain-specific reinforcement training successfully transfers into broader logical reasoning enhancements.
> 3. Limitations and Attribution: Despite strong performance, we observed that improvements were less significant on specific tasks (e.g., AGIEval-Gaokao-Chinese and LogiQA). As noted in literature **[1]**, the reasoning generalization derived from cross-domain RL is often uneven and is partially constrained by the base model's pre-training familiarity with specific domains.
> Given that we did not conduct any fine-tuning on the remaining domains, no new knowledge was introduced to the model. Therefore, these performance gains can be attributed to enhanced reasoning capabilities, which enable the model to more effectively utilize its internal pre-existing knowledge. While extensive research has focused on reasoning generalization across multiple domains, our training phase relies on a significantly narrower source domain. Consequently, our work may offer novel insights into the mechanisms of reasoning generalization.

---

> > ### Author Response · Authors · 2025-11-21
> >
> > ### OOD Benchmarks
> > | Model                         | AGIEval | MMLU   | MMLU-Pro | MedExpQA | MedMCQA | MedQA  |
> > |------------------------------|---------|--------|----------|----------|---------|--------|
> > | Qwen2.5-7B-Instruct          | 0.6486  | 0.7091 | 0.4552   | 0.5260   | 0.5319  | 0.6540 |
> > | Morpheus-7B         | 0.6628  | 0.7862 | 0.5795   | 0.5820   | 0.5929  | 0.7200 |
> > | Qwen2.5-14B-Instruct         | 0.7013  | 0.8010 | 0.6271   | 0.6420   | 0.5929  | 0.7346 |
> > | Morpheus-14B        | 0.7468  | 0.8434 | 0.6665   | 0.7520   | 0.6608  | 0.7984 |
> > | Qwen2.5-32B-Instruct         | 0.7394  | 0.8284 | 0.6715   | 0.7460   | 0.6278  | 0.7796 |
> >
> > ### MedExpQA
> >
> > | Model                         | English | French | Italian | Spanish |
> > |------------------------------|---------|--------|---------|---------|
> > | Qwen2.5-7B-Instruct          | 0.600   | 0.512  | 0.512   | 0.480   |
> > | Morpheus-7B         | 0.632   | 0.608  | 0.464   | 0.624   |
> > | Qwen2.5-14B-Instruct         | 0.712   | 0.576  | 0.632   | 0.648   |
> > | Morpheus-14B       | 0.736   | 0.728  | 0.776   | 0.768   |
> > | Qwen2.5-32B-Instruct         | 0.784   | 0.768  | 0.712   | 0.720   |
> >
> > ### MedQA
> >
> > | Model | Mainland | Taiwan | US |
> > |---|---|---|---|
> > | Qwen2.5-7B-Instruct | 0.797 | 0.660 | 0.505 |
> > | Morpheus-7B | 0.795 | 0.709 | 0.656 |
> > | Qwen2.5-14B-Instruct | 0.816 | 0.745 | 0.643 |
> > | Morpheus-14B | 0.835 | 0.815 | 0.746 |
> > | Qwen2.5-32B-Instruct | 0.853 | 0.804 | 0.682 |
> >
> > ### AGIeval
> >
> > | Model | AQuA-RAT | Gaokao | JEC | LSAT | LogiQA | SAT |
> > |---|---|---|---|---|---|---|
> > | Qwen2.5-7B-Instruct | 0.622 | 0.740 | 0.563 | 0.476 | 0.528 | 0.722 |
> > | Morpheus-7B | 0.721 | 0.737 | 0.575 | 0.489 | 0.530 | 0.766 |
> > | Qwen2.5-14B-Instruct | 0.752 | 0.767 | 0.601 | 0.533 | 0.621 | 0.798 |
> > | Morpheus-14B | 0.772 | 0.817 | 0.670 | 0.585 | 0.648 | 0.830 |
> > | Qwen2.5-32B-Instruct | 0.744 | 0.817 | 0.663 | 0.576 | 0.652 | 0.804 |
> >
> > ### MMLU
> >
> > | Model | Medical | college | high_school | other | professional |
> > |---|---|---|---|---|---|
> > | Qwen2.5-7B-Instruct | 0.728 | 0.595 | 0.773 | 0.703 | 0.581 |
> > | Morpheus-7B | 0.782 | 0.734 | 0.859 | 0.773 | 0.673 |
> > | Qwen2.5-14B-Instruct | 0.797 | 0.724 | 0.864 | 0.796 | 0.693 |
> > | Morpheus-14B | 0.842 | 0.791 | 0.905 | 0.834 | 0.741 |
> > | Qwen2.5-32B-Instruct | 0.827 | 0.801 | 0.896 | 0.812 | 0.732 |
> >
> > ### MMLU-Pro
> >
> > | Model                       | Health | Biology | Business | Chemistry | Computer Science | Economics | Engineering | History | Law   | Math  | Other | Philosophy | Physics | Psychology |
> > |-----------------------------|---------|---------|----------|-----------|-------------------|-----------|-------------|---------|-------|-------|--------|------------|---------|------------|
> > | Qwen2.5-7B-Instruct         | 0.502   | 0.695   | 0.459    | 0.413     | 0.400             | 0.556     | 0.366       | 0.417   | 0.299 | 0.486 | 0.432  | 0.361     | 0.400   | 0.589      |
> > | Morpheus-7B | 0.630   | 0.738   | 0.649    | 0.572     | 0.637             | 0.679     | 0.339       | 0.533   | 0.326 | 0.759 | 0.556  | 0.453     | 0.590   | 0.654      |
> > | Qwen2.5-14B-Instruct        | 0.643   | 0.787   | 0.710    | 0.657     | 0.649             | 0.719     | 0.490       | 0.541   | 0.341 | 0.778 | 0.590  | 0.529     | 0.657   | 0.691      |
> > | Morpheus-14B  | 0.703   | 0.827   | 0.693    | 0.627     | 0.749             | 0.778     | 0.396       | 0.625   | 0.445 | 0.810 | 0.672  | 0.629     | 0.642   | 0.734      |
> > | Qwen2.5-32B-Instruct        | 0.686   | 0.829   | 0.734    | 0.691     | 0.698             | 0.751     | 0.550       | 0.583   | 0.406 | 0.799 | 0.648  | 0.581     | 0.719   | 0.727      |
> >
> >
> >
> > ### Reference
> > [1] Cheng, Zhoujun, et al. "Revisiting Reinforcement Learning for LLM Reasoning from A Cross-Domain Perspective." arXiv preprint arXiv:2506.14965 (2025).

---

> > > ### Comment · Reviewer_eXNB · 2025-11-21
> > >
> > > I would like to thank the authors for providing detailed responses, which addressed most of my concerns. I would raise my score.

---

> > > > ### Author Response · Authors · 2025-11-26
> > > >
> > > > We sincerely appreciate your positive feedback and the valuable effort you have invested in refining our manuscript.

---

### Official Review · Reviewer_EdCA · 2025-10-31

**Soundness:** 3
**Presentation:** 4
**Contribution:** 2
**Rating:** 6
**Confidence:** 4

**Summary:**

The paper introduces a comprehensive suite of artifacts for the medical domain of anesthesiology, including benchmarks, a specialized corpus, and a anesthesiology-aligned model. The authors detail their end-to-end pipeline, covering pre-training data collection, benchmark creation, and the continuous training of a base model on the resources. The proposed benchmark is used to benchmark current SoTA medical LLMs, as well as open-weight and frontier models.

**Strengths:**

- The authors have conducted a substantial amount of work, delivering all the necessary sources and artifacts for the medical domain of anesthesiology, a field that is currently underserved by existing resources.

- The paper presents a solid methodology for assessing both contamination and originality across all proposed resources, ensuring the integrity of their contributions.

**Weaknesses:**

**[W1]** While the authors acknowledge that anesthesiology is underrepresented in current medical NLP resources and provide a setup that uses both reasoning and factual knowedge, they do not offer a clear, tangible motivation for how these resources will benefit medical-aligned LLMs overall (i.e. examples illustrating how these resources could be applied in real-world scenarios).


**[W2]** The methodology for benchmark creation and human verification is insufficiently detailed. Although their hybrid approach (LLMs + human verification) is reasonable, the paper does not specify:
- The number of human annotators involved per sample
- The instructions or rubric used for verification
- Annotator agreement rates
- How quality is ensured for data not verified by humans
These missing details make it difficult to assess the reliability and validity of the benchmarks.

**[W3]** The paper does not discuss how alignment in the anesthesiology domain affects performance on other downstream medical tasks (e.g., MedQA) or general tasks (e.g., MMLU).

**[W4]** The pre-training and SFT data are likely to be used in continuous pretraining of medical-aligned models that need to perform well across multiple medical domains. However, the authors do not provide a realistic CPT scenario demonstrating how their data would be used in such a context. Even a hypothetical example would help clarify the potential impact and application of their resources.

**[W5]** The use of anesthesiology-related documents from Fineweb for ontinuous pretraining involves replaying existing data rather than introducing new resources. As a result, this aspect of the contribution is less substantial compared to the creation of the main benchmark.

**Questions:**

Please refer to the Weaknesses section above.

---

> ### Author Response · Authors · 2025-11-21
> **Response to Reviewer EdCA:**
>
> Thank you for your constructive feedback. We have carefully considered your suggestions and have outlined our responses below.
>
> ---
>
> ## Response to Weakness 1: Motivation and Example
>
> We sincerely appreciate and acknowledge the reviewer's suggestion to clarify our motivation and real-world applicability. We agree that addressing "underrepresentation" alone is insufficient. Our core motivation is that anesthesiology serves not just as a sub-specialty, but as an ideal testbed to cultivate and evaluate the "System 2" reasoning capabilities required for robust medical AI.
>
> Our motivation and the practical utility of these resources are outlined as follows:
> 1. Motivation: Real-world anesthesiology and perioperative scenarios require models to go beyond simple "fact retrieval" and possess deep reasoning capabilities to handle conflicts and dynamic changes. For example:
> 	1. Resolving Therapeutic Conflicts: Models must often resolve conflicting treatment requirements. For instance, in a patient with "aortic stenosis complicated by renal failure," an LLM cannot simply retrieve standard treatment protocols; it must carefully weigh conflicting contraindications. Here, the LLM needs to focus on balancing circulatory stability while simultaneously protecting kidney function.
>     2. Real-World Time Sensitivity: In scenarios such as surgical "Malignant Hyperthermia," correct decision-making depends largely on the precise timing and sequence of interventions, not merely on symptom recognition.
>
> For more examples regarding conflicting constraints and context sensitivity in perioperative scenarios, please refer to the Case Study in Appendix E of the paper.
>
> The examples above demonstrate that possessing only general medical knowledge is insufficient to meet real-world challenges. Due to its close association with perioperative care and multiple specialties, anesthesiology becomes a critical scenario for training and testing a model's ability to flexibly apply cross-disciplinary knowledge and understand clinical context.
>
> 2. Examples: In summary, AnesSuite contributes to the broader field of medical LLM alignment through the following avenues:
> 	1. Enhancing Cross-Disciplinary Integration: Given the inherently interdisciplinary nature of anesthesiology, training on our collected resources—including the perioperative corpus (AnesCorpus) and Q&A pairs based on anesthesiology literature (AnesQA)—encourages models to synthesize multidisciplinary knowledge effectively.
> 	2. Reasoning Enhancement: The dataset designed for RLVR training within anesthesiology and perioperative contexts (AnesR1) can further elevate a model's ability to perform "reasoning under constraints." This capability is transferable, improving the reliability and safety of LLMs in other high-risk fields, thereby promoting the overall alignment of LLMs with complex medical needs.
>
> Furthermore, our anesthesiology reasoning benchmark (AnesBench) can be utilized alongside other medical benchmarks to evaluate the knowledge recall and reasoning capabilities of general medical models across multiple specialties.
>
> ---
>
> ## Response to Weakness 2:  Human Verification
>
> We appreciate your constructive feedback regarding our manual verification methods. We concur that manual verification is a crucial factor in validating the quality of a benchmark. As demonstrated in **Appendix A.3**, our screening process covered over 60% of the benchmark (approximately 5,000 questions), resulting in an error rate of 0.34%. We hereby provide a detailed elaboration on the manual review methodology employed for this benchmark data:
>
> 1. Review Workflow: To strike a balance between verification scale and depth, we established a quality control protocol involving sampling and a two-tiered review process. Since our benchmark is not LLM-generated, we recruited two postgraduate students with relevant professional backgrounds to serve as trained domain annotators. They are able to verify the completeness, formatting, and fundamental correctness of the questions.
> 	1. Phase 1: The two annotators independently reviewed the assigned samples, tasked with flagging any sample suspected of quality issues.
> 	2. Phase 2: All samples identified as "suspected anomalies" in the first stage entered a second stage of joint review.

---

> ### Author Response · Authors · 2025-11-21
>
> 2. Validation Protocols: To guide review operations, we established evaluation guidelines across three dimensions:
> 	1. Completeness Check: Verified that the question stems and options were intact and that numerical values of key clinical indicators were not truncated. Any instance where missing critical information rendered the question unanswerable was classified as a completeness error.
> 	2. Correctness & Scientific Validity: Conducted verification of standard answers and fact-checking of medical content. Samples were flagged if the standard answer was incorrect, the question contained logical contradictions, or if it included obsolete medical concepts.
> 	3. Formatting & Structural Compliance: Checked for data corruption (garbled text), escape character errors, and other formatting issues to ensure the data met the standards for both machine processing and human readability.
> 3.  Consistency Verification
> We implemented a consistency confirmation process for anomalous samples.
> We required the two annotators to conduct a joint review of all potential noise samples identified in the first stage. Any discrepancies or uncertainties were resolved by reaching a consensus based on anesthesiology guidelines and established clinical standards. This process yielded a final error rate of **0.34% (17 items)** within the sampled subset.
> 4. Quality Assurance for Unreviewed Data
> Our validation covered 60% of the benchmark, exhibiting a low error rate. Statistically, the consistent distribution and the minimal error rate allow us to reasonably extrapolate this quality conclusion to the remaining unreviewed data. This implies that the error rate in the unreviewed data also remains at an acceptable level.
>
> Despite this high expected quality, we remain committed to long-term maintenance. Upon open-sourcing the resources, we will simultaneously launch an online errata to continuously collect community feedback and release periodic version updates, ensuring the long-term validity.
>
> ---
>
> ## Response to Weakness 3:  OOD Performance
>
> We appreciate you and reviewer eXNB raising this important question regarding the model's generalization capabilities. We systematically evaluated Morpheus on six OOD benchmarks. These covered both general medical contexts (MedQA, MedMCQA, MedExpQA) and general domains (MMLU, MMLU-Pro, AGIEval).
> Our experimental results indicate that SFT+GRPO fine-tuning on anesthesia data not only avoided overfitting but significantly enhanced the model's general reasoning capabilities. We will update the manuscript to include the following key findings:
> 1. Out-of-Distribution Medical Tasks: Morpheus-7B/14B outperformed the Qwen2.5-Instruct baseline of the same scale across all three general medical benchmarks. In some cases, they even surpassed the performance of 32B models. Furthermore, the model exhibited significant cross-lingual transfer capabilities, showing consistent improvements across French as well as Spanish and Traditional Chinese.
> 2. Generalization to General Tasks: On general benchmarks such as MMLU and AGIEval, the model's overall performance also surpassed the baselines. This suggests that domain-specific reinforcement training successfully transfers into broader logical reasoning enhancements.
> 3. Limitations and Attribution: Despite strong performance, we observed that improvements were less significant on specific tasks (e.g., AGIEval-Gaokao-Chinese and LogiQA). As noted in literature **[1]**, the reasoning generalization derived from cross-domain RL is often uneven and is partially constrained by the base model's pre-training familiarity with specific domains.
> Given that we did not conduct any fine-tuning on the remaining domains, no new knowledge was introduced to the model. Therefore, these performance gains can be attributed to enhanced reasoning capabilities, which enable the model to more effectively utilize its internal pre-existing knowledge. While extensive research has focused on reasoning generalization across multiple domains, our training phase relies on a significantly narrower source domain. Consequently, our work may offer novel insights into the mechanisms of reasoning generalization.
>
> For AGIeval, the evaluation is limited to its MCQ subsets, whereas all other benchmarks are evaluated in their entirety. Please refer to **Appendix G.1** for detailed evaluation specifications.

---

> > ### Author Response · Authors · 2025-11-21
> >
> > ### OOD Benchmarks
> > | Model                         | AGIEval | MMLU   | MMLU-Pro | MedExpQA | MedMCQA | MedQA  |
> > |------------------------------|---------|--------|----------|----------|---------|--------|
> > | Qwen2.5-7B-Instruct          | 0.6486  | 0.7091 | 0.4552   | 0.5260   | 0.5319  | 0.6540 |
> > | Morpheus-7B         | 0.6628  | 0.7862 | 0.5795   | 0.5820   | 0.5929  | 0.7200 |
> > | Qwen2.5-14B-Instruct         | 0.7013  | 0.8010 | 0.6271   | 0.6420   | 0.5929  | 0.7346 |
> > | Morpheus-14B        | 0.7468  | 0.8434 | 0.6665   | 0.7520   | 0.6608  | 0.7984 |
> > | Qwen2.5-32B-Instruct         | 0.7394  | 0.8284 | 0.6715   | 0.7460   | 0.6278  | 0.7796 |
> >
> > ### MedExpQA
> >
> > | Model                         | English | French | Italian | Spanish |
> > |------------------------------|---------|--------|---------|---------|
> > | Qwen2.5-7B-Instruct          | 0.600   | 0.512  | 0.512   | 0.480   |
> > | Morpheus-7B         | 0.632   | 0.608  | 0.464   | 0.624   |
> > | Qwen2.5-14B-Instruct         | 0.712   | 0.576  | 0.632   | 0.648   |
> > | Morpheus-14B       | 0.736   | 0.728  | 0.776   | 0.768   |
> > | Qwen2.5-32B-Instruct         | 0.784   | 0.768  | 0.712   | 0.720   |
> >
> > ### MedQA
> >
> > | Model | Mainland | Taiwan | US |
> > |---|---|---|---|
> > | Qwen2.5-7B-Instruct | 0.797 | 0.660 | 0.505 |
> > | Morpheus-7B | 0.795 | 0.709 | 0.656 |
> > | Qwen2.5-14B-Instruct | 0.816 | 0.745 | 0.643 |
> > | Morpheus-14B | 0.835 | 0.815 | 0.746 |
> > | Qwen2.5-32B-Instruct | 0.853 | 0.804 | 0.682 |
> >
> > ### AGIeval
> >
> > | Model | AQuA-RAT | Gaokao | JEC | LSAT | LogiQA | SAT |
> > |---|---|---|---|---|---|---|
> > | Qwen2.5-7B-Instruct | 0.622 | 0.740 | 0.563 | 0.476 | 0.528 | 0.722 |
> > | Morpheus-7B | 0.721 | 0.737 | 0.575 | 0.489 | 0.530 | 0.766 |
> > | Qwen2.5-14B-Instruct | 0.752 | 0.767 | 0.601 | 0.533 | 0.621 | 0.798 |
> > | Morpheus-14B | 0.772 | 0.817 | 0.670 | 0.585 | 0.648 | 0.830 |
> > | Qwen2.5-32B-Instruct | 0.744 | 0.817 | 0.663 | 0.576 | 0.652 | 0.804 |
> >
> > ### MMLU
> >
> > | Model | Medical | college | high_school | other | professional |
> > |---|---|---|---|---|---|
> > | Qwen2.5-7B-Instruct | 0.728 | 0.595 | 0.773 | 0.703 | 0.581 |
> > | Morpheus-7B | 0.782 | 0.734 | 0.859 | 0.773 | 0.673 |
> > | Qwen2.5-14B-Instruct | 0.797 | 0.724 | 0.864 | 0.796 | 0.693 |
> > | Morpheus-14B | 0.842 | 0.791 | 0.905 | 0.834 | 0.741 |
> > | Qwen2.5-32B-Instruct | 0.827 | 0.801 | 0.896 | 0.812 | 0.732 |
> >
> > ### MMLU-Pro
> >
> > | Model                       | Health | Biology | Business | Chemistry | Computer Science | Economics | Engineering | History | Law   | Math  | Other | Philosophy | Physics | Psychology |
> > |-----------------------------|---------|---------|----------|-----------|-------------------|-----------|-------------|---------|-------|-------|--------|------------|---------|------------|
> > | Qwen2.5-7B-Instruct         | 0.502   | 0.695   | 0.459    | 0.413     | 0.400             | 0.556     | 0.366       | 0.417   | 0.299 | 0.486 | 0.432  | 0.361     | 0.400   | 0.589      |
> > | Morpheus-7B | 0.630   | 0.738   | 0.649    | 0.572     | 0.637             | 0.679     | 0.339       | 0.533   | 0.326 | 0.759 | 0.556  | 0.453     | 0.590   | 0.654      |
> > | Qwen2.5-14B-Instruct        | 0.643   | 0.787   | 0.710    | 0.657     | 0.649             | 0.719     | 0.490       | 0.541   | 0.341 | 0.778 | 0.590  | 0.529     | 0.657   | 0.691      |
> > | Morpheus-14B  | 0.703   | 0.827   | 0.693    | 0.627     | 0.749             | 0.778     | 0.396       | 0.625   | 0.445 | 0.810 | 0.672  | 0.629     | 0.642   | 0.734      |
> > | Qwen2.5-32B-Instruct        | 0.686   | 0.829   | 0.734    | 0.691     | 0.698             | 0.751     | 0.550       | 0.583   | 0.406 | 0.799 | 0.648  | 0.581     | 0.719   | 0.727      |

---

> > > ### Author Response · Authors · 2025-11-21
> > >
> > > ## Response to Weakness 4: Potential Impact and Application
> > >
> > > We appreciate your professional and insightful comments. We concur with your perspective on utilizing both pre-training data and SFT data for the CPT of medically aligned models. This approach has the potential to grant the model a higher performance ceiling and enable rapid convergence on downstream tasks **[2, 3, 4]**.
> > >
> > > To elucidate the potential impact and application value of the AnesSuite resources, we have constructed a hypothetical scenario demonstrating how to leverage AnesCorpus (related to anesthesiology and the perioperative period) and AnesR1 (anesthesia reasoning data) to augment the capabilities of general medical alignment models.
> > >
> > > For an LLM that already possesses foundational medical alignment capabilities, our suggested CPT strategy is designed as follows:
> > > 1. Data Mixture Strategy: Although existing models possess a degree of multi-specialty knowledge, anesthesiology and perioperative data are often under-represented in general corpora. Therefore, AnesCorpus can serve as a pivotal supplement. When mixed with general medical domain data to form the foundational training corpus $D_C$, it broadens the model's knowledge scope regarding perioperative management and interdisciplinary physiological regulation.
> > > 2. Injection of Reasoning Capabilities: During the annealing phase of CPT, we propose employing a lightweight Instruction Synthesizer **[2]** to synthesize a subset of "instruction-response" pairs based on $D_C$, denoted as $D_I$. The final training data ratio for this phase can be set to approximately 80% $D_C$ and 20% $D_I$, with the additional incorporation of AnesR1 (which contains long Chain-of-Thought data).
> > > In summary, as emphasized in our response to Weakness 1, anesthesiology is characterized by its intrinsic multidisciplinary intersection and strong reasoning-driven nature. Integrating AnesCorpus and AnesR1 into the CPT process of medical alignment models serves a dual purpose: it not only mitigates the scarcity of domain-specific data while preventing the loss of model diversity caused by significant domain shifts, but also infuses complex clinical reasoning capabilities early into the pre-training phase, ultimately elevating the model's overall proficiency in medical decision-making.
> > >
> > > ## Response to Weakness5: Contributions by AnesCorpus
> > >
> > > We sincerely appreciate the reviewer's insightful feedback. We fully understand your concerns regarding the derivation of the AnesCorpus dataset from FineWeb and the magnitude of its contribution. We would like to clarify its unique value through the following two perspectives:
> > > 1. Contribution to Domain Alignment While AnesCorpus is constructed based on FineWeb, its core contribution lies in achieving a alignment from general-purpose text to the specialized domain of anesthesiology. We have refocused the dispersed, general text within FineWeb into the specific domain of "Anesthesiology and Perioperative Medicine". As demonstrated in Section 5.2, we empirically confirmed the practical value of this dataset.
> > > 2. Contribution to a Controllable Experimental Environment:
> > > 	1. Two-stage decontamination protocol ensures strict independence between the training and test sets, establishing a reliable benchmark environment for future research.
> > > 	2. This decontamination process incurs significant overhead. For instance, FineWeb contains over 15 trillion text tokens, requiring in excess of 44TB of disk space. Consequently, the resource consumption associated with data transmission, storage, processing, and deduplication against the test set is non-negligible.
> > >
> > > Ultimately, we wish to emphasize that AnesCorpus constitutes just one component of the broader AnesSuite, and our work represents a systemic contribution designed to provide a comprehensive infrastructure for the field.
> > >
> > > ### Reference
> > > [1] Cheng, Zhoujun, et al. "Revisiting Reinforcement Learning for LLM Reasoning from A Cross-Domain Perspective." arXiv preprint arXiv:2506.14965 (2025).
> > >
> > > [2] Cheng, Daixuan, et al. "Instruction Pre-Training: Language Models are Supervised Multitask Learners." EMNLP, 2024.
> > >
> > > [3] Akter, Syeda Nahida, et al. "Front-Loading Reasoning: The Synergy between Pretraining and Post-Training Data." arXiv preprint arXiv:2510.03264 (2025).
> > >
> > > [4] Hu, Shengding, et al. "MiniCPM: Unveiling the Potential of Small Language Models with Scalable Training Strategies." COLM, 2024.

---

> ### Author Response · Authors · 2025-11-26
>
> As the discussion period progresses, we sincerely hope that our rebuttal has effectively addressed your concerns. We value your insights and would welcome any further feedback you may have. Thank you for your time and consideration.

---

### Official Review · Reviewer_utdx · 2025-11-01

**Soundness:** 3
**Presentation:** 3
**Contribution:** 2
**Rating:** 4
**Confidence:** 4

**Summary:**

The paper introduces AnesSuite, a dataset and benchmark collection designed to test and improve LLM reasoning in anesthesiology. AnesBench provides evaluation tasks that cover factual knowledge and complex decision making in English and Chinese. The authors also release three training datasets that target continued pretraining, supervised fine tuning, and reinforcement learning with verifiable rewards. Using these resources, they build a baseline model named Morpheus and run a broad set of experiments against strong public models. The analysis examines how language transfer, training data choices, and training methods affect performance on anesthesiology reasoning.

**Strengths:**

1.  The study makes a clear and timely observation that evaluation should reflect realistic clinical settings, and it identifies contextual complexity as a common bottleneck for reasoning across tasks.

2. The three-tier benchmark design and the linked training resources form a coherent pipeline, and the Morpheus baselines with ablations provide useful evidence on which choices lead to gains.

**Weaknesses:**

1. I appreciate the careful build of a bilingual, cognitively tiered benchmark and the broad empirical sweep, yet the main novelty lies in the dataset suite rather than a method or algorithm that is central to ICLR. The focus on anesthesiology is narrow for a broad ML venue and is also narrow even within medical or clinical tracks, which limits general use.

2. The dataset and benchmark draw mainly on public exams and textbooks and are text only. They do not include images, multimodal clinical data, or real clinical or PHI data. Existing resources already support assessment of medical knowledge, such as MedQA, MedMCQA, and PubMedQA. Clinically grounded datasets that capture real decision-making and multimodal signals are more directly relevant to patient care. The contribution is limited to that, not a direct link to clinically actionable use.


3. I am concerned about data quality. On AnesBench, SFT on the proposed dataset does not beat the Qwen2.5-Instruct baselines: Morpheus 7B with SFT only averages 54 percent compared with 59 percent for the 7B baseline, and Morpheus 14B with SFT only averages 57 percent compared with 64 percent for the 14B baseline. Most of the gains appear after the GRPO stage, to 63 percent and 69 percent, and that is modest. SFT slightly improves the English subset but drops the Chinese subset by a large margin, which points to issues with data quality, data balance, or objective alignment. Train and test sit in the same domain due to vertical fine-tuning. One would usually expect larger SFT gains in that case, so the results are underwhelming.

**Questions:**

Please see the question about weakness above.

---

> ### Author Response · Authors · 2025-11-21
> **Response to Reviewer utdx:**
>
> We are grateful for the constructive criticism, which has helped us significantly improve the quality of our manuscript. Our point-by-point responses are provided below.
>
> ---
>
> ## Response to Weakness 1:  Research Scope and General Use
>
> We acknowledge the reviewer's two primary concerns: (1) the work is characterized by dataset contributions rather than methodological or algorithmic novelty, questioning its alignment with ICLR; and (2) the focus on anesthesiology appears too narrow, potentially limiting generalizability. We clarify as follows:
>
> 1. Alignment with ICLR’s Scope: Domain-specific data and fine-grained benchmarks are critical for advancing model performance and probing capability boundaries **[1-2]** . AnesSuite offers a comprehensive reasoning infrastructure. It integrates a bilingual, cognitively stratified, and verifiable evaluation suite with aligned training resources (CPT/SFT/RLVR) and systematic empirical analysis. This "infrastructure + empirical rigor" approach is critical for advancing the field and constitutes a suitable topic for the ICLR.
> 2. Our contribution is not limited to enhancing domain-specific anesthesiological knowledge. Instead, we aim to leverage the high-pressure, reasoning-intensive scenarios of anesthesiology to refine core LLM reasoning capabilities. Unlike static, single-disease tasks that rely heavily on pattern recognition, anesthesiology highly demands complex reasoning to make clinical decisions in dynamic environments. The robust reasoning mechanisms acquired in this extreme environment effectively transfer to broader domains. Our additional experimental results (please refer to our response to Weakness 2 raised by reviewer eXNB) confirm that models fine-tuned exclusively on anesthesiology data achieved significant gains on general medical (e.g., MedQA) and general cognitive (e.g., MMLU-Pro, AGIeval) benchmarks. This demonstrates that training in "strong reasoning scenarios" yields generalized benefits across diverse tasks, rather than resulting in domain overfitting.
>
> ---
>
> ## Response to Weakness 2:  Positioning and Multimodal Integration
>
> We thank the reviewer for the opportunity to clarify the positioning and clinical relevance of AnesSuite.
>
> Unlike MedQA and PubMedQA, which primarily focus on general medical knowledge or simple reasoning, AnesSuite is specifically designed to address the high-stakes, systematic decision-making challenges inherent in anesthesiology. As shown in Table 3, the significant performance gaps of state-of-the-art models on system 2 tasks directly substantiate the necessity of this benchmark.
>
> Furthermore, we posit that textual reasoning is a logical prerequisite for clinical multimodal reasoning. If the model cannot perform accurate reasoning based on precise text descriptions, it is expected to fail when processing noisy real-world multimodal data. We fully recognize the value of integrating real-world data and are actively expanding AnesSuite into a multimodal benchmark by incorporating datasets such as MIMIC-IV and MOVER. We view the current study as the cornerstone of this future work.
>
> ---
>
> ## Response to Weakness 3: AnesR1 Data Quality
>
> We thank the reviewer for raising this important question. We take your concerns regarding data quality very seriously and address them through the following two points:
>
> 1. Training Strategy: Following the established "SFT+RLVR" paradigm, we positioned the SFT phase as a cold start prior to RL, designed primarily for domain alignment via knowledge injection and the acquisition of basic reasoning patterns. Research suggests that maintaining high policy entropy before reinforcement learning is critical for maximizing the performance upper bound **[3]**. To prevent the model from prematurely converging to local optima or overfitting to specific output patterns, we intentionally limited the SFT to just 100 steps. The temporary performance drop you noted likely reflects the "alignment tax" associated with this format adaptation. However, as demonstrated in Table 4, the subsequent GRPO stage successfully unlocked the model's potential, yielding significant gains over the baseline.
> 2. Supplementary Experiments: To rigorously determine whether the performance decline stemmed from data quality or model-specific sensitivity, we conducted control experiments using Llama-3.1-8B,  Gemma2-2B and Gemma3-4B on the same AnesR1 dataset (extended to 200 SFT steps). These experiments showed consistent improvements across both English and Chinese tasks for both models. This directly demonstrates that the AnesR1 dataset contains high-quality, effective domain knowledge. Furthermore, it suggests that the performance drop you observed is likely attributable to model-specific sensitivity rather than issues regarding data quality or data balance.

---

> ### Author Response · Authors · 2025-11-21
>
> | Model                | AnesBench-English | AnesBench-Chinese | Avg. |
> |------------------------------|-------------------|-------------------|------|
> | Llama-3.1-8B-Instruct         | 0.54              | 0.48              | 0.51 |
> | Llama-3.1-8B(AnesR1)    | 0.60              | 0.51              | 0.55 |
> | Gemma2-2B-Instruct           | 0.40              | 0.34              | 0.37 |
> | Gemma2-2B(AnesR1)       | 0.45              | 0.41              | 0.43 |
> | Gemma3-4B-Instruct           | 0.32              | 0.40              | 0.36 |
> | Gemma3-4B(AnesR1)       | 0.36              | 0.43              | 0.39 |
>
> We sincerely appreciate your feedback and concern. We will incorporate these additional experimental results into the revised manuscript to provide further reference for readers(all modifications in the paper are highlighted in blue.).
>
> ### Reference
>
> [1] Xie, Yunfei, et al. "Medtrinity-25m: A large-scale multimodal dataset with multigranular annotations for medicine." ICLR, 2025.
>
> [2] Saxon, Michael, et al. "Benchmarks as microscopes: A call for model metrology." COLM, 2024.
>
> [3] Cui, Ganqu, et al. "The entropy mechanism of reinforcement learning for reasoning language models." arXiv preprint arXiv:2505.22617 (2025).

---

> ### Author Response · Authors · 2025-11-26
>
> With the discussion period now halfway through, we would like to gently follow up to ensure that our responses and revisions have adequately addressed the concerns you raised. We value your constructive feedback and remain fully available to provide any further clarification needed. Thank you for your time and guidance so far.

---

### Official Review · Reviewer_rYyW · 2025-11-01

**Soundness:** 2
**Presentation:** 2
**Contribution:** 2
**Rating:** 6
**Confidence:** 2

**Summary:**

The paper introduces AnesSuite, a suite aimed at assessing and improving anesthesiology reasoning in LLMs. It comprises:
(1) AnesBench — a bilingual (EN/ZH) multiple‑choice benchmark (7,972 items) stratified by cognitive demand into System 1 (factual), System 1.x (hybrid), and System 2 (complex decision making).
(2) AnesCorpus — ~2.4M anesthesia‑related documents (1.8M EN + 0.6M ZH) for continued pretraining (CPT), filtered from FineWeb/Chinese‑FineWeb with keyword rules and a two‑stage decontamination (n‑gram screen + LCS>64 removal) to reduce overlap with the benchmark.
(3) AnesQA — 20,713 English QA pairs produced via a two‑model LLM pipeline (Llama‑3.3‑70B to generate questions; Qwen‑2.5‑72B to filter/answer), with manual spot‑checks and regex filtering to excise 119 flawed items.
(4) AnesR1 — 10,287 verifiable MCQs (EN+ZH) with chain‑of‑thought (CoT) traces generated by DeepSeek‑R1 and kept via rejection sampling only when the final choice is correct. Using these resources, the authors train Morpheus‑7B/14B (Qwen‑2.5 inits) with SFT + GRPO; they report improvements over the instruction‑tuned bases and parity with larger baselines on AnesBench.

**Strengths:**

- First-of-it-kind domain‑specific reasoning suite for anesthesiology that separates fast recall from complex decision‑making (System 1→2) and offers training corpora aligned with CPT/SFT/RLVR stages. Prior work either subsumed anesthesiology under broad medical categories or focused on factual recall; this suite targets decision‑making explicitly;

- Bilingual design with explicit experiments on language transfer. The ablation shows CPT on English‑heavy corpora can help EN while hurting ZH, highlighting data‑language balance as a first‑order concern in medical domains;

- Reasoning‑centric baselines (Morpheus) showing that limited RL with verifiable rewards (GRPO) plus CoT data can close the gap to much larger models on this domain.

**Weaknesses:**

1. External validity of System‑2 constructed from exam/textbook‑style abstractions rather than real perioperative trajectories; this may under‑represent key anesthetic complexities (e.g., dynamic vitals, ventilator settings, lab time‑series).

2. Leakage analysis scope: the permutation‑confidence heuristic over 500 items is informative but indirect; it cannot rule out contamination in proprietary models, nor leakage via near‑duplicates beyond LCS>64.

3. GRPO is described at a high level; the exact verifiable reward signal (e.g., answer‑only vs. additional structure) and stability/variance aren’t deeply analyzed.

**Questions:**

1. For AnesR1, did clinicians evaluate a subset of DeepSeek‑R1 chains for medical correctness beyond the final answer? Any evidence that RL on these chains avoids entrenching subtle misconceptions?

2. What exact reward(s) were used in GRPO—answer correctness only, or also intermediate checks (e.g., vital‑sign logic)? How robust are results to reward misspecification?

---

> ### Author Response · Authors · 2025-11-21
> **Response to Reviewer rYyW:**
>
> We appreciate your valuable comments and constructive feedback. Please find our detailed responses below.
>
> ---
> ## Response to Weaknesses 1: External Validity of System‑2 Subset
>
> We share your view on the complexity of anesthesia. To address your concerns regarding external validity, we offer the following clarifications:
>
> 1. Decoupling Perception and Reasoning: Clinical competence comprises two distinct stages: Perception (accurately reading and interpreting raw multimodal data) and Reasoning (synthesizing decisions based on extracted information). AnesSuite focuses specifically on the latter. While text is inherently abstract, our scenarios includethe core clinical elements required for complex clinical judgment (as seen in Example 1). This allows us to present complex, conflicting signals within a concise format to effectively test the model's reasoning capabilities.
> 2. Evaluation Efficiency in "Long-Tail" Scenarios: Abstract scenarios serve as an effective proxy for real clinical situations that are rare and high-stakes. As illustrated in Example 2 (sudden abdominal compartment syndrome during gastroschisis repair leading to hemodynamic collapse), achieving scale for such rare events is challenging when relying solely on real-world clinical logs.
> 3. Roadmap for Multimodal Integration: To bridge the gap in evaluating the perception layer, we are upgrading our benchmark using real-world medical databases (e.g., MOVER, MIMIC-IV) via a "semi-automated construction + human verification" pipeline. Specifically:
>     1. Slot Filling: We plan to identify segments in existing MCQs that involve data interpretation (e.g., "A {age}-year-old, ASA {class} patient... with MAP at {MAP_value} mmHg for {duration} minutes...").
>     2. Data Injection: We will employ expert models to replace these textual descriptions with real multimodal data segments from the aforementioned databases.
>     3. Comprehensive Assessment: This will force the model to first perceive real perioperative time-series or imaging data before applying medical knowledge for reasoning.
>
> ```
> Example-1：
> ID : 53e144d6-b8f9-5760-bf4d-2ea90fd3e755
> Question:"A 33-year-old primigravid woman with myasthenia gravis, well-controlled with pyridostigmine, is in labor with the cervix dilated 7 cm. She has a headache and feels very nervous. Blood pressure is 160/115 mmHg, she has 3+ pitting edema, and urinalysis shows 4+ protein. Appropriate management of her labor should include"
> "A": "lumbar epidural block with bupivacaine 8 ml of 0.5% solution",
> "B": "chlorpromazine 2.5 mg administered intravenously",
> "C": "avoidance of narcotics",
> "D": "lumbar epidural block with 2-chloroprocaine 8 ml of 3% solution",
> "E": "chlorpromazine 10 mg administered intramuscularly"
> Example-2：
> ID ：373a5c3d-f9b5-5f87-853c-0b5f13d560e5
> Question："An infant becomes cyanotic and hypotensive after ventilatory pressures increase to 45 cm H2O during closure of the abdominal wall at the end of a gastroschisis repair. The most appropriate next step is to"
> "A": "administer a muscle relaxant",
> "B": "administer 5% dextrose in lactated Ringer's solution 10 ml/kg",
> "C": "change the endotracheal tube",
> "D": "ask the surgeon to insert a chest tube",
> "E": "ask the surgeon to reopen the abdomen"
> ```
>
> ---
>
> ## Response to Weaknesses 2: Leakage Analysis
>
> We appreciate the reviewer's insightful comments.
>
> Regarding contamination in proprietary models : We acknowledge the inherent difficulty in rigorously verifying data contamination in proprietary models due to the unavailability of training data and output distributions. The permutation-based method requires access to token probabilities, which restricts its application to open-source models. While label-only membership inference attacks exist for black-box models, current techniques generally exhibit low reliability (e.g., AUC < 0.6). Therefore, we employed the permutation-based method to validate the open-source models, which aligns with our MCQ data format.
>
> Regarding "near-duplicates" and the LCS threshold: We argue that our decontamination protocol is sufficiently rigorous to effectively exclude significant near-duplicates. We utilized a 64-character LCS threshold, which is significantly stricter than established standards in the medical domain; for instance, Med-PaLM 2 employed a 512-character threshold. A 64-character overlap typically corresponds to only a single short sentence or a specific medical phrase. By filtering at such fine granularity, we aggressively removed not only exact copies but also substantial partial overlaps and "near-duplicate content," thereby ensuring the robustness and credibility of the experimental results based on AnesSuite.

---

> > ### Author Response · Authors · 2025-11-21
> >
> > ## Response to Weaknesses 3: Details of GRPO
> >
> > We thank the reviewer for the comment regarding the implementation details and stability analysis of our GRPO training. We have revised the manuscript to include a detailed description of the reward mechanism and a comprehensive stability analysis in **Appendix C.2**. Our response is as follows:
> >
> > The reward is binary ($r \in \{0, 1\}$). A reward of $1$ is assigned solely when the model's final answer matches the ground truth; otherwise, the reward is $0$. To facilitate this, we employ robust regular expressions (matching patterns including, but not limited to, "Answer is" or "Answer:") to strictly parse the final answer from the model's output.
> >
> > We agree that providing more detailed results and analysis is beneficial for a deeper discussion. In the revised Appendix (revisions to the manuscript have been marked in blue.), we have included Figure 9 and Figure 10 , which illustrates the reward trajectories for both Morpheus-7B and Morpheus-14B (including three independent runs for Morpheus-7B). The curves exhibit consistent trends throughout the training steps, showing no signs of collapse or divergence.
> >
> > ---
> >
> > ## Response to Question 1
> >
> > Following the construction of AnesR1, we invited senior clinical anesthesiologists to evaluate several CoT reasoning paths within the dataset. The results indicated that the majority of these reasoning chains were robust. However, the experts also observed instances that were verbose or contained minor discrepancies due to regional differences in clinical practice (e.g., variations in first-line medication recommendations across different guidelines). Nevertheless, they deemed these to be acceptable clinical variations rather than fundamental errors in medical logic or fact.
> > Regarding whether RL on these chains can mitigate misconceptions, we observed the following positive evidence:
> > 1. As shown in Tab.4, the Morpheus exhibited significant performance improvements after both SFT and GRPO training.
> > 2. We also invited clinical anesthesiologists to analyze the cases presented in Appendix E, and they concluded that the reasoning process referenced the most up-to-date guidelines and demonstrated superior quality.
> >
> > Nevertheless, the evidence presented above does not fully guarantee that RL on these chains can entirely avoid misconceptions. To reduce or even completely eliminate misconceptions, future work must incorporate Process Reward Models or Retrieval-Augmented Generation techniques.
> >
> > ## Response to Question 2
> >
> > In our GRPO implementation, we exclusively utilized answer correctness as the reward. However, we strongly agree with your insight regarding the importance of intermediate verification. The final answer to MCQs acts merely as a proxy reward during training; the true objective is to ensure that every step of the model's output embodies accurate medical knowledge and sound reasoning processes.
> >
> > Therefore, incorporating intermediate checks, such as the 'vital-sign logic', is crucial to mitigate reward misspecification. However, implementing such granular verification typically requires expensive expert-level annotation to train a process reward model.
> >
> > Nevertheless, as demonstrated in Table 4 and our response to weakness 2 raised by reviewer eXNB, GRPO robustly enhances model performance across multiple domains, even without explicit measures to prevent reward misspecification. Furthermore, the stable reward curves presented in our response to Weakness 3 provide partial evidence of the method's robustness against potential reward misspecification.

---

> > > ### Comment · Reviewer_rYyW · 2025-11-21
> > >
> > > Thank you for your detailed rebuttal. All my previous concerns have been well addressed and I improve my score accordingly. I really appreciate it the comprehensive, large-scale training resources and the rigorous evaluation design. Thus I vote for accept.

---

> > > > ### Author Response · Authors · 2025-11-26
> > > >
> > > > We are grateful for your encouraging comments and the time you dedicated to helping us improve our work.

---

### Author Response · Authors · 2025-12-01
**Summary of Reviews and Rebuttal for the Area Chair**

Dear Area Chair,

We sincerely appreciate the time and effort you have dedicated to managing our submission. We also extend our gratitude to the reviewers for their constructive feedback. Below, we summarize the reviews and detail how we have addressed the reviewers' concerns during the rebuttal phase.

Overall, reviewers rYyW and eXNB confirmed that our responses resolved their concerns and subsequently raised their scores (6→8). Reviewer EdCA provided a positive initial review (though he/she has not yet responded to our rebuttal). Regarding Reviewer utdx, who expressed concerns about AnesR1 data quality, we not only clarified our training strategy but also conducted supplementary experiments on three additional models to validate the dataset's effectiveness. We remain confident that these results effectively address Reviewer utdx's concerns.

Our work received recognition from the reviewers in several key areas:
- Comprehensive and coherent resources (Reviewers rYyW, utdx, EdCA, eXNB)
- Extensive experimentation, including baselines and key insights (Reviewers rYyW, utdx, eXNB)
- Sound benchmark design (hierarchical & bilingual) (Reviewers rYyW, utdx)
- Solid decontamination protocols (Reviewer EdCA)
- High-quality visualizations (Reviewer eXNB)

Regarding the constructive criticism and weaknesses raised, we have addressed the **common concerns** as follows:

- Non-multimodal nature of the benchmark (Reviewer rYyW W1 & Reviewer utdx W2): We clarified the benchmark's focus on reasoning and explained the advantages of a text-based benchmark for long-tail scenarios. However, acknowledging the importance of multimodal integration, we provided a detailed and feasible roadmap for upgrading AnesBench to a multimodal benchmark using existing real-world resources.
- Contribution to broader fields (Reviewer utdx W1, Reviewer EdCA W3/4, & Reviewer eXNB W2/Q1): We clarified the nature of anesthesiology as a reasoning-driven, interdisciplinary field, making these resources valuable for broader applications. Following suggestions from Reviewers EdCA and eXNB, we added results from three general medical benchmarks and three general-domain benchmarks (OOD). The results show that our reasoning baseline, Morpheus, performs surprisingly well on these OOD tasks, offering potential insights into reasoning generalization mechanisms. Furthermore, we outlined a hypothetical CPT example based on AnesSuite to demonstrate its utility for general medical alignment models.

In addition to these common points, we addressed the **specific concerns** of each reviewer:

- Reviewer rYyW (Rating: 6→8):
    - Contamination Analysis (W2): Regarding contamination assessment for proprietary models, we explained that label-only methods are unreliable for strict verification. We adopted a heuristic approach aligned with our data format and clarified that our filtering threshold is stricter than the standards used in established medical LLMs.
    - GRPO Details (W3): We provided detailed definitions of the rewards used and presented training curves for Morpheus-7B/14B (reporting the mean and standard deviation across repeated runs for the 7B model), demonstrating training stability.

- Reviewer utdx (Rating: 4):
    - AnesR1 Data Quality (W3): The reviewer noted a performance drop in the Chinese subset after SFT. We clarified that this was due to intentionally limiting the training steps to maintain high policy entropy, rather than data quality issues. We further conducted supplementary experiments on Llama-3.1-8B, Gemma2-2B, and Gemma3-4B, none of which showed this performance drop, suggesting the phenomenon was likely due to model-specific sensitivity rather than the dataset itself.

- Reviewer EdCA (Rating: 6):
    - Motivation (W1): We clarified the unique value of anesthesiology as a testbed for "System 2" reasoning under high-stakes, conflicting constraints.
    - Human Verification (W2): We detailed our two-stage quality control protocol and highlighted the extremely low error rate (0.34%), which covered over 60% of the benchmark (~5,000 items).
    - AnesCorpus Contribution (W5): We emphasized the corpus's core value in aligning general text to a specific domain, as well as its systematic contribution as part of AnesSuite. Furthermore, we highlighted the significant effort required for rigorous decontamination to ensure a controlled experimental environment.

- Reviewer eXNB (Rating: 6→8):
    - Benchmark Challenge (W1): We pointed out that although performance on "System 1" tasks boosted the overall scores, even SOTA models struggle with complex anesthesiology reasoning and are far from ready for clinical application, which underscores the difficulty of the benchmark. Additionally, through a supplementary evaluation based on open-ended generation, we verified that the current MCQ format remains a valid and reliable proxy for assessing complex anesthesiology reasoning.

---

> ### Author Response · Authors · 2025-12-01
>
> In summary, during the discussion phase, we conducted the following key supplementary experiments:
>
> - We expanded the OOD evaluation by testing Morpheus on six additional benchmarks, covering both general medical tasks and general-domain tasks. The results show that our training strategy substantially enhances the model’s general reasoning ability, rather than merely leading to source-domain overfitting.
> - We validated the quality of the AnesR1 dataset by conducting supplementary training on different model families (Llama-3.1-8B, Gemma2-2B, Gemma3-4B). The experiments show consistent performance improvements, confirming the effectiveness of the dataset and ruling out data-quality issues.
> - We evaluated the training stability of GRPO by running independent repeated experiments and reporting the mean and standard deviation of the reward curves, demonstrating that our method exhibits consistent convergence with no signs of collapse.
>
> We highly value the reviewers’ thoughtful feedback, and the above supplementary experiments, as well as their corresponding updates, have been highlighted in blue in the revised manuscript.
>
> We once again thank the AC, SAC, PC, and all reviewers for their time and professional guidance. We hope that this work can make a meaningful contribution to the medical AI community.
>
> Sincerely,
>
> Authors of Submission 3903

---

### Meta-Review · Area_Chair_YFxp · 2025-12-25

**Summary:**

The paper proposes AnesSuite, a comprehensive suite for evaluating and improving LLM reasoning in anesthesiology. It consists of several parts: (1) AnesBench, a bilingual, cognitively-tiered benchmark (System-1 to System-2 reasoning); (2) AnesCorpus, domain-filtered pretraining corpus with rigorous decontamination; (3) AnesQA & AnesR1, supervised and RL-ready reasoning datasets; (4) Morpheus models, baseline 7B/14B models trained via SFT + GRPO. The work aims to study reasoning under high-stakes perioperative decision-making and to examine whether such training transfers beyond anesthesiology.

Reviewers mentioned several strengths. (1) It is the first systematic anesthesiology reasoning suite. (2) Clear cognitive stratification (System-1 / 1.x / 2). (3) Bilingual design with language-transfer analysis. (4) Extensive experimentation and ablations. (5) Stable RL training analysis and reward curves.

Two reviewers initially raised concerns but stated that the rebuttal addressed them and increased their scores. The rebuttal provided clarification on benchmark motivation, reasoning focus, design choices, and evaluation methodology, and added several supplementary experiments.

There remain some limitations. The benchmark currently focuses on textual reasoning and has not yet incorporated multimodal or clinical-trajectory components, and reinforcement learning still relies on answer-level rewards rather than process-level verification. However, these are positioned as natural next steps rather than omissions that weaken the present contribution.

**Reviewer Concerns:**

Another concern related to the benchmark’s reliance on text-based and exam-style abstractions rather than multimodal or real clinical trajectories. The authors explained that the benchmark intentionally isolates the reasoning layer rather than the perception layer, arguing that text-based abstraction provides an efficient way to probe high-stakes and rare perioperative scenarios. They also outlined a concrete plan to extend the benchmark toward multimodal evaluation using existing clinical databases. This clarification makes the current scope more conceptually grounded.

Another major concern centered on the performance drop observed after SFT on the Chinese subset, which raised questions about data quality and training alignment. The rebuttal clarified that the SFT stage was intentionally kept short to preserve policy entropy before reinforcement learning and that the purpose of SFT here was domain alignment rather than final optimization.

A further point raised by reviewers involved potential in-distribution bias in the evaluation of Morpheus. In response, the authors added out-of-distribution experiments on both medical and general reasoning benchmarks, showing that models trained on anesthesiology reasoning data do not overfit but achieve improvements on broader tasks. While the generalization gains are not uniform across all categories, the results support the authors’ central claim that training in reasoning-intensive contexts can enhance cross-domain reasoning.

**Reviewer Scores:**

Two reviewers initially raised concerns but increased their scores later.

---

### Decision · Program_Chairs · 2026-01-26

Accept (Poster)